



# Measurement Report: Strong Valley Wind Events during the International Collaborative Experiment - PyeongChang 2018 Olympic and Paralympic Winter Games Project

Paul Joe[1], Gyuwon Lee[2], and Kwonil Kim[2]

[1]Environment and Climate Change Canada (retired), 4905 Dufferin St., Toronto, Ontario, Canada M3H 5T4
[2]Department of Astronomy and Atmospheric Sciences, Center for Atmospheric REmote sensing (CARE), Kyungpook National University, Daegu, Republic of Korea

**Correspondence:** Paul Joe (paul.joe@hotmail.ca)

**Abstract.** Strong gusty wind events were responsible for some of the poor performances of competitors and resulted in schedule changes during the PyeongChang 2018 Olympic and Paralympic Winter Games. Three events at two venues were investigated to document and articulate the wind forecasting and nowcasting challenges. Upper air analysis showed that the Games were dominated by northwesterly synoptic flow. Froude and Reynolds number analyses indicated that vortex shedding or wake
turbulence were the dominant mechanisms in the lee of the mountains where the free-style competitions were conducted. Three types of wind data (10 and 1 min averages plus 1 minute maximums) from automatic weather stations that were reported every minute were analyzed using advanced techniques (Hovmueller, wavelet and eigen analysis frequency estimation). For the two days of Event 1, the conditions were well mixed throughout the day and night. For the other events, diurnal variations were observed with a stable atmosphere at night, well mixed in the afternoon and with 2-4 hour transition periods in the morning
and evenings. Turbulence was best portrayed using wavelet analysis and vortex shedding was best portrayed using the eigen analysis frequency estimation method. The latter revealed dominant frequencies, presumably associated with vortex shedding with periodicities of 20 to 90 minutes. Nowcast implications are discussed.

## 1  Introduction

Strong gusty wind events were responsible for the poor performance of competitors and resulted in schedule changes during
the PyeongChang Winter Olympics Games. Training and qualification events were cancelled and the rules of competition were altered to accommodate safe conditions for the competitors, to conduct fair competitions, to optimize television viewership, to hold medal final events daily, and other considerations.

These events were conducted in complex terrain which are subject to synoptic and to local and diurnal valley effects, all of which can generate small scale turbulent flows. The multi-scale, and in particular the small scale nature of winds, was evident
as gusts affected competitors (scheduled to compete every 1-2 minutes apart) in different ways leading to unfair competition conditions. For example, head wind gusts slowed the speed of the participants in the women's slope style event (12 Feb 2018) to the extent that some competitors did not have sufficient speed to launch off the jumps or land them safely or securely (WSS,


2021). Some competitors abandoned their attempts effectively ending their competition. No competitor was able to complete the two run finals without falling. In the men's half pipe event (22 Feb 2018), cross-winds would re-direct the arc of the jumps

either out of or into the half pipe causing loss of speed or control with many competitors having to abandon their runs.

For consistency, the term "field of play" will be used as a generic term to describe where the competition is conducted in lieu of other common terms such as "race course", "ski or competition slope" or the "run". It is noteworthy that the field of play is extremely small by normal operational forecast scales. In the case of the freestyle events, the distance from top to bottom may only be a few hundred meters in length and for the Alpine speed events, the distance was about 2 kilometres along the slope

with less than 1km in vertical extent.

The rules of competition define the nature of the event (slope steepness, difficulty), the manner in which it is conducted (including the number of training runs, qualification procedures) and also govern safety and fairness of the competition (often in non-technical meteorological terms). During the Olympics, there is considerable leeway by the organizers in interpretation of the competition rules with the ability to adjust the schedule.

For the longer and faster Alpine events (e.g. the downhill), there can be significant differences in the weather along the race course. It can be foggy (in-cloud) at the top, snowing in the middle and raining below that (Mo et al., 2013). Safety may be restricted by visibility and the general rule is that the competitor must be able to see two turns or gates ahead. For shorter and slower Alpine events such as the slalom, this can be of the order of 5 to 10 meters but for the speed events, this can be 100 to 200 meters. Gusty conditions can significantly affect ski jumpers or aerialists providing significant advantage or disadvantage. An

individual competition occurs over approximately a 90 minute period and consistency of conditions is important to fairness. In lieu of a calm 90 minute period, judges may decide to conduct the competition between gusty periods, shorten the race course or alter the competition rules to shorten the event time. The competition actually starts several days before the finals, as training sessions are under strict control of the judges to ensure fairness so that the competitors must have an equal number of training runs and to ensure safety.

So, not only nowcasts at a microscale but short-term forecasts are required to conduct an event. Unlike normal competitions that have a limited time window (a weekend) to be conducted where if conditions are not appropriate, the competitions are often cancelled. During the Olympics, events can be rescheduled within a one- or even two-week window and so longer term forecasts for very specific conditions and locations are also required.

The fine scale patterns of winds in complex terrain have been the subject of recent research studies. Windward-side studies

have examined the relationship between blocked/unblocked flows and microphysics on the location and intensity of precipitation (Stoelinga et al., 2003; Steiner et al., 2003; Theriault et al., 2012). Complex wind flows in valleys, basins, plain-foothills, lee-side, gaps have been the subject of several recent major field campaigns and demonstrated the wide variety of localized wind flows that may arise (Fernando et al., 2019; Fernando et al., 2015; Whiteman, 2000; Tsai et al., 2021; Park et al., 2021). Teakles et al., (2013) describe the impact and forecast issues facing Olympic forecasters even during seemingly benign sunny

days with early morning drainage flows transitioning to upslope flow due to day time heating.

Though, several projects have addressed these issues and the aggregate knowledge has been shared (Horel et al., 2002; Joe et al., 2010; Isaac et al., 2014; Kiktev et al., 2017), each project is unique and pose considerable challenges of multi-scale,





specificity in terms of weather elements and fluidity in service requirements. This is beyond the scope of routine forecast services and even the current generation of high resolution numerical weather prediction models.

The International Collaborative Experiment for PyeongChang 2018 Olympic and Paralympic Winter Games (ICE-POP 2018) was an international research development project led by the Korean Meteorological Administration (KMA) under the auspices of the World Meteorological Organization (WMO). KMA was responsible for providing weather services to the PyeongChang Olympic and Paralympic Winter Games. The primary objective of ICE-POP 2018 was to advance the science and improve the parameterization of winter microphysics in high resolution numerical weather prediction models by leverag-

ing international expertise and by symbiotic deployment, enhancement and use of advanced and high resolutions observations and to support Olympic weather services (Gehring et al., 2020; Lee et al., 2021). ICE-POP 2018 had an extensive experimental campaign with enhanced surface and upper air observations, research radars, microphysical aircraft and ship borne observations, a variety of high resolution numerical weather prediction models and analyses systems (Lee et al., 2021). For this contribution, detailed automatic weather station wind observations reported every minute were available and extensively used

at the Alpine (Jeongson, JS) and freestyle venues (Bokwang, BOK or Phoenix Park). Upper air observations (as frequently as every 6 hours)were available from the MOO station located less than 20 km away.

Events were postponed or rescheduled based on existing conditions but also on the forecast. The competitors experienced performance influencing wind effects at time scales of less than a minute and at very fine spatial scales (meters, for example, in-between jumps). The goal of this contribution is to investigate the issues of nowcasting winter events for the Olympics (an

engaged end-user) and the adequacy, efficacy or utility of the one minute wind data to meet the challenges. The objectives are to (i) analyze the wind data to investigate what the data are able to reveal using advanced techniques not commonly available to forecasters (ii) from a hindsight perspective, investigate the events with respect to the scheduling decisions, (iii) to articulate and further document the challenge of winter Olympic nowcasting and (iv) to share insights.

The paper is organized as follows: (i) The venues and fields of play, the selected events and available observations are briefly

described, (ii) followed by the results from various advanced analyses (Hovmueller, wavelet and frequency spectral estimation), (iii) then by a discussion of the insights and (iv) a summary.

## 2   Project Background/Setup

The ICE-POP 2018 project study area was in the north east of South Korea (Figure 1 and is approximately 100 km x 100 km square. The host city was Gangneung which is located on the coastal plain. To the immediate west is a north-south mountain

chain with the ridge line about 15-25 km to the west and parallel to the coast and approximately 1100 meters in height. West of this mountain ridge is a high plain (30-55 km) of 500-600 m altitude surrounded by mountains.

To make the analysis tractable, this investigation focused on three events at two of the venues, Bokwang (BOK) and Jeong-song (JS) located on the western edge of the project area (Table 3). BOK is the site of the freestyle events where the competitors are exposed to gusty strong winds when they execute jumps up to 10 or more meters into the air. BOK lies along a broad rela-

tively open north-south slope with the highest peak Peak B at 1236 m just to the northwest of the competitions (600-1100m).

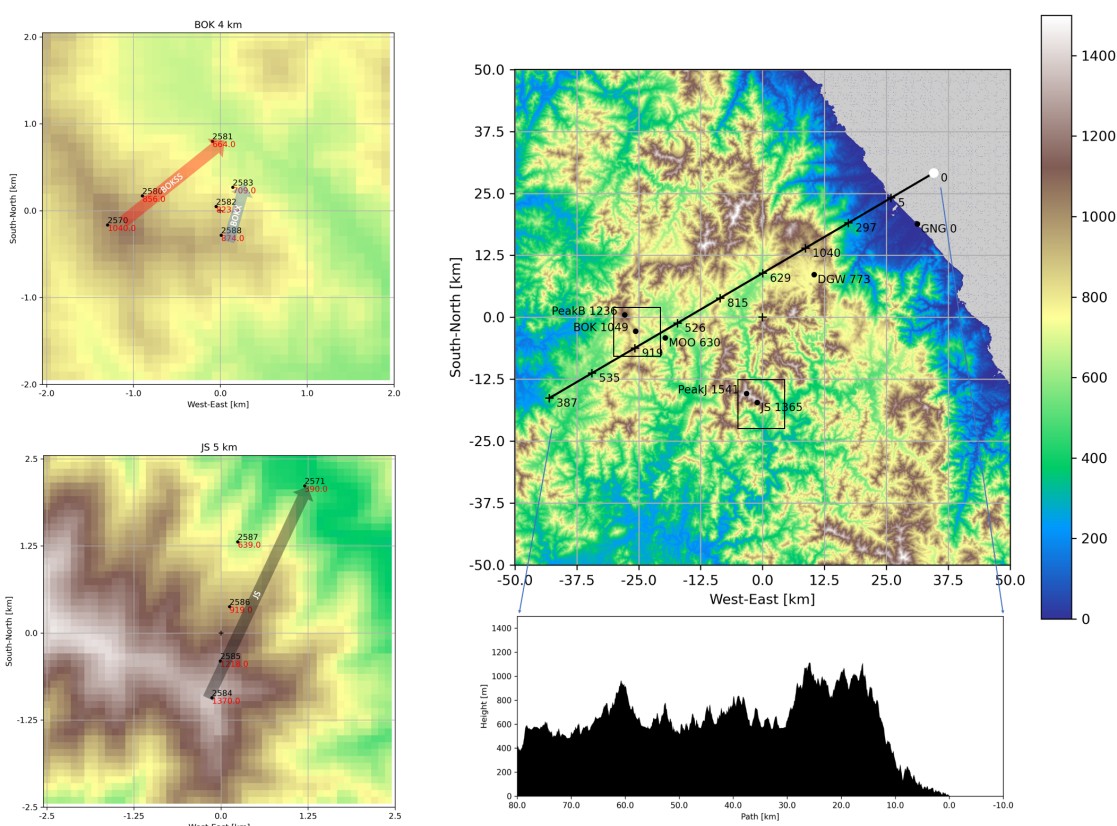

**Figure 1.** The ICE-POP 2018 project area is an approximately 100km x 100km area in the north east of South Korea. Major venues and upper air stations (DGW and MOO) are indicated and the elevation profile (bottom right) shows the complexity of the terrain and that the dominant feature is the coastal mountains that rises sharply from the sea of Japan. The inset figures (on the left) show details of the terrain (30 m resolution from SRTM03) and the location of the automatic weather stations at the Bokwang (BOK) and Jeongsong (JS) Olympic venues which are the foci of this paper. BOKX, BOKSS and JS are transects through the weather stations along the fields of play.





**Table 1.** Venues

| Location | Latitude | Longitude | Altitude [m] | Comment |
|---|---|---|---|---|
| JS | 37.444256 | 128.588241 | 1365 | Top of Alpine Downhill |
| BOK | 37.573815 | 128.310003 | 1049 | Top off Freestyle |
| GNG | 37.769219 | 128.955637 | 0 | Coastal Land Sea Boundary |
| DGW | 37.677331 | 128.718825 | 773 | Upper Air Site |
| MOO | 37.562000 | 128.377600 | 630 | Upper Air Site |
| PeakB | 37.603420 | 128.284398 | 1236 | Highest Peak north of Freestyle |
| PeakJ | 37.460971 | 128.563685 | 1541 | Highest Peak north of Jeongson |

**Table 2.** Wind Events

| Event | Start Date | End Date | Event | Description |
|---|---|---|---|---|
| 1 | Feb 11 | Feb 12 | Women's Slope Style | Qualification cancelled on Feb 11 due to strong winds, Finals held on Feb 12 when winds were even stronger. |
| 2 | Feb 15 | Feb 16 | Transition | For comparison. |
| 3 | Feb 21 | Feb 23 | Alpine Event | Strong winds predicted, events moved to avoid Feb 22. |
| Winter Season | Dec 1 2017 | Mar 31 2018 | 4 month period. | For context. |

The BOK venue was further partitioned into two transects, BOKX and BOKSS, which are only about a kilometer apart, that are the site of the ski cross (XC) and slope style (SS) events. The transects are separated by groves of trees. JS is the site of the downhill and super-G alpine speed events. In contrast to BOK, the highest peak (Peak J at 1541) is located near the race course that is constructed along a narrow east-west avalanche chute.

Three events during the Olympic period (9-25 Feb 2018, with extra days bracketing the opening and closing dates ) were selected for study (Table 2) with the objective of documenting the conditions and implications for nowcasting.

In the first event, gusty winds on the afternoon of Sunday 11 Feb 2018 resulted in cancelling qualification runs for the women's slope style competition based on the observations and nowcasts. The competition rules were then altered to eliminate qualification runs altogether and for all competitors to go directly to a two-run final competition on Monday 12 Feb 2018. While the short-range forecast was for gusty winds on Monday, the hope was that the gustiness would be less. Observations, nowcasts and test runs would be conducted in late morning of Monday to determine if the competition could be held in the early afternoon. It should be noted that freestyle events, in general, are relatively new events for the Olympics and experience by all





involved, including competition organizers, is limited. In hindsight, the conditions on Monday were worse than the previous day.

The second two day event (15-16 Feb 2018) had calmer wind conditions with diurnal effects were evident (see below) and was included for comparison purposes.

The third event (21-23 Feb 2018) was a three-day event near the end of the Olympics. Based on the short-term forecast from early in the week, winds on Thursday 22 Feb 2018 were forecast to be strong. Given the recent experience and impact, the competition organizers cancelled the competition for Thursday and moved events both forward to Wednesday and backward
to Friday to avoid the predicted strong wind conditions for Alpine ski events.

For contextual purposes, analysis of the the entire winter (defined to be 1 Dec 2017 to 31 March 2018) is presented in a summary fashion. It should be noted that there were strong wind events at other venues such as the ski jump and in the host city of Gangneung during and prior to the Olympics.

Full details of the ICE-POP 2018 instrumentation are described elsewhere (Lee et al., 2021). The primary observations used
in this study were provided by automatic weather stations (Vaisala WXT520) collecting basic meteorological data at 1 minute resolution located on the mountain slopes and six hourly upper air soundings at the nearby MOO station. The location of the stations are presented in Figure 1 and Table 1. The AWS winds were reported every minute but processed as 10 minute (WS10) and 1 minute (WS1) running averages as well as the maximum in the past minute (WSS). Considering the complex nature of the terrain, It should be noted that not all AWS stations were equivalently sited in "open" terrain but located near and hence
reflected (and interpreted as) the conditions of the field of play.

## 3   Wind Analysis

### 3.1   Wind Time Series

Figure 2 shows the wind trace for all three events for the weather stations at the top of the two venues (BOK and JS) to introduce the automatic weather station (AWS) observations. Temperature, pressure and humidity (not shown) were also measured as
ten minute running averages. For Events 1 and 3, there are significant fluctuations of the later two winds compared to the ten minute averages. This is not observed in Event 2 indicating the strong but much less gusty winds.

Figure 3 show a two hour segment of the data at Jeonson (9 - 11 KST 13 Februrary 2018) illustrating the fluctuation differences between the 3 wind types that is not shown in Figure 2 due to the scale/extent of the data presentation. Note that all times are in Korean Standard Time (KST; UTC+9) for convenience.

### 3.2   Upper Air Analysis

Climatologically, there are three persistent synoptic features that affect the project area/Korean Peninsula in winter (Kim et al., 2021). The passages of low pressure systems, in particular, warm lows, cause significant snowfalls in the western sides of the mountain ridges and in the coastal Gangneung area. A Siberian high brings dry cold northerly air and an easterly flow

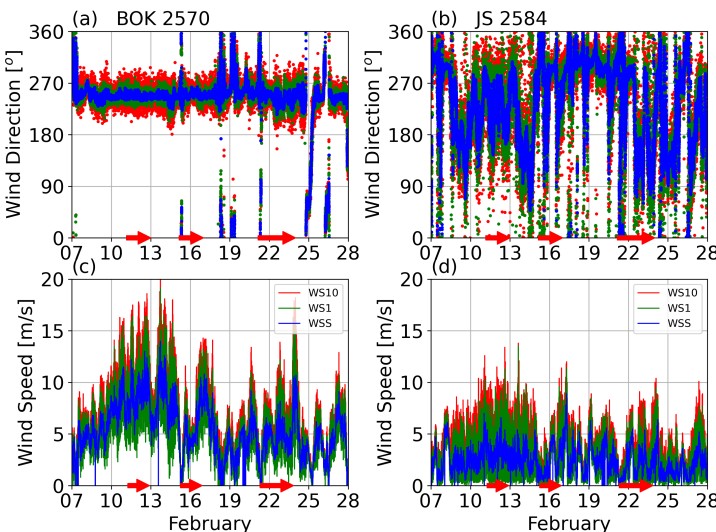

**Figure 2.** Time series of the wind at the top of Bokwang and Jeongsong for the 3 events investigated. Winds were reported every minute consisting of 10 minute average (WS10), 1 minute average (WS1) and maximum within the latest 1 minute (WSS). The red arrows at the bottom indicate the date-time of the events investigated (Events 1-3, left to right, see Table 2).

from the Sea of Japan can bring warm moist air to the Gangneung area leading to intense coastal precipitation. Figure 4 shows

wind speed and direction time-height plots from the upper air soundings from MOO station located within 40 km of the two venues (see Figure 1). For essentially, the entire winter period and certainly during the Olympic period, the dominant flow was a northerly/northwesterly cold and strong flow from the Siberian High with a periodic southerly and south-westerly flow due to a low pressure passage. In the period between the Olympics and Paralympics, a low level eastern flow pattern, led to significant precipitation events.

The 700 mb winds, Froude and Reynolds Number time traces are shown in the lower three panels of Figure 4. The strength of the 700 mb winds have been previously used to distinguish synoptic from local wind regimes due either to complex terrain/valley or diurnal effects(Whiteman and Doran, 1993). Similarly, the Froude number and the Reynolds number are used to indicate whether the upstream air will pass over (Colle, 2004) or around elevated terrain and indicate if flow separation is expected in the lee of the mountains, respectively.

The 700 mb wind strength is virtually greater than 7 m/s for all times, the Froude number is greater than 1 and the Reynolds number is greater than $10^5$ for almost the entire winter period. These all indicate that synoptic flows are predominant and

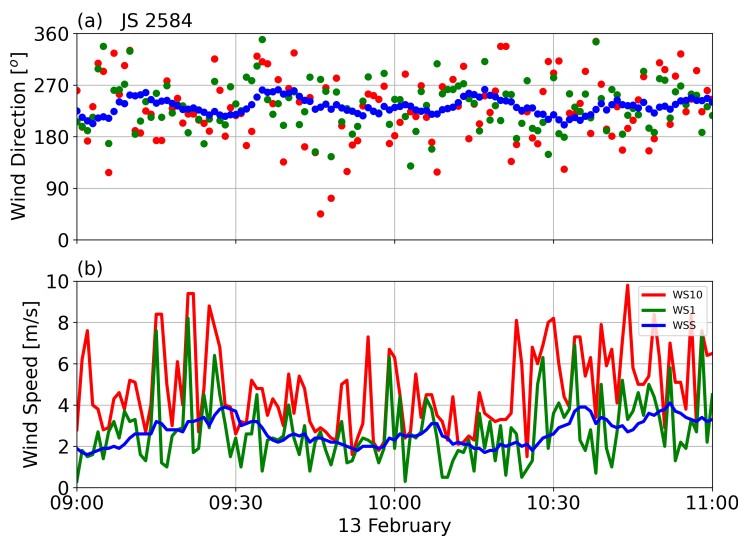

**Figure 3.** Two hour sequence of the winds at the top of Jeongson for 13 February 2018 9 to 11 UTC illustrating the greater variation of the WS1 and WSS compared to the WS10 winds.

that the local lee side winds at BOK and JS (and elsewhere) will be due to mechanical turbulence (vortex shedding or wave turbulence) rather than due to diurnal, plain-slope transitions though there are periods of local diurnal processes are evident (Teakles et al., 2013; Whiteman and Doran, 1993; Fernando et al., 2015, Fernando et al., 2019). The interpretation of high

Reynolds number is imprecise. Flow around idealized a circular cylinder (Chang, 1970) indicate that low (1 to 10), moderate ($10$ to $10^5$) and high ($> 10^5$) Reynolds numbers, correspond to bound, vortex shedding and wake turbulence regimes, respectively. Mountains with irregular edges, shapes and roughness would expect to exhibit flow separation and turbulent wakes at lower values when compared to smooth blunt two dimensional objects.

### 3.3   Hovmueller Analysis

The data can be succinctly presented using Hovmueller diagrams which are altitude-time diagrams. Three transects were identified along the ski cross (BOKX), slope style (BOKSS) and downhill (JS) competition venues (Figure 1, Table 3). Hovmueller diagrams of potential temperature and wind were constructed to investigate the thermodynamic and wind structure of the air along the slope. The winds were resolved in co- and cross-slope directions with positive being upslope and right-ward direc-

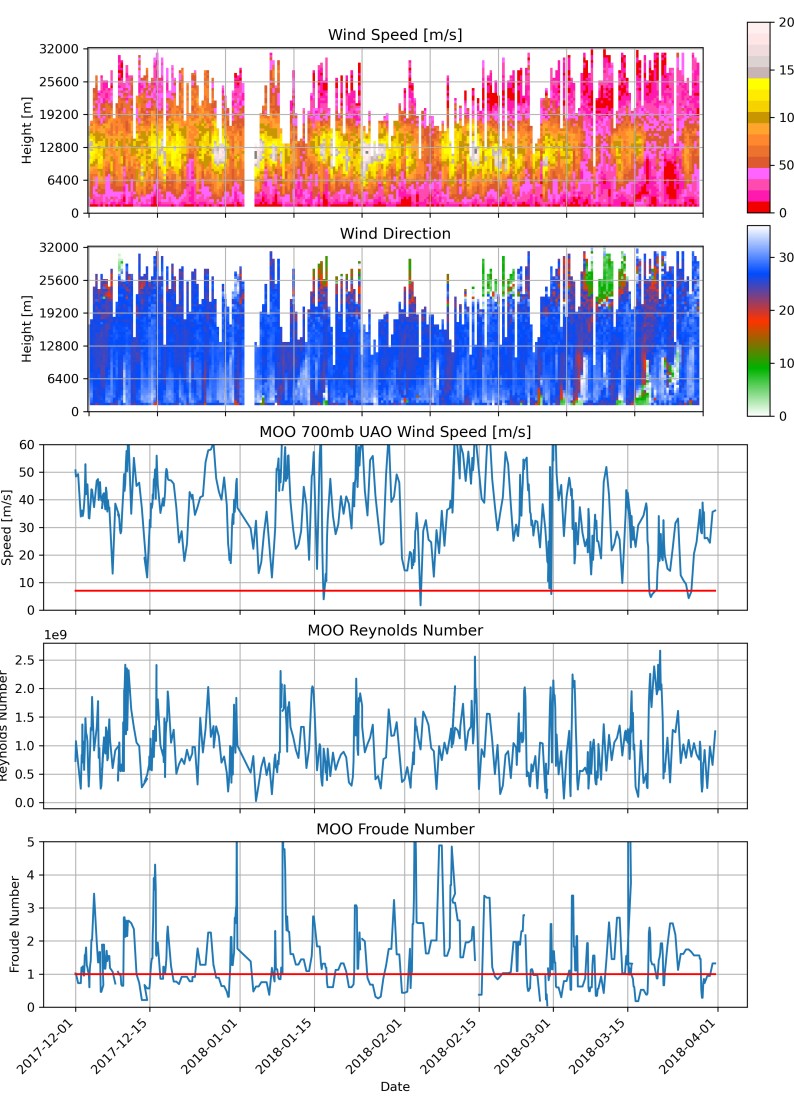

**Figure 4.** Wind speed and direction from MOO Upper Air Station for the entire winter season (1 Dec 2017 to 31 Mar 2018) showing that the predominant winds are northwesterly. The MOO station is located within 40 km of BOK and JS venues and in the lee of the mountains. Froude number and Reynolds are crudely estimated to indicate that wind flow separation over the mountains.





**Table 3.** Transects

| Venue Transect | Station Number | Longitude | Latitude | Altitude [m] | Distance [m] | Slope Drop [m] | Slope Direction [°] | Incline [°] |
|---|---|---|---|---|---|---|---|---|
| BOKX | 2570 | 128.3080533 | 37.5754252 | 1040 | 0 | 0 | 0 | 0.0 |
| BOKX | 2580 | 128.3126975 | 37.5785269 | 856 | 536 | 184 | 230 | 19.0 |
| BOKX | 2581 | 128.3220392 | 37.5843297 | 664 | 1047 | 192 | 232 | 10.4 |
| BOKX | Summary | | | | 1582 | 376 | 231 | 13.4 |
| BOKSS | 2588 | 128.3232113 | 37.5743463 | 874 | 0 | 0 | 0 | 0.0 |
| BOKSS | 2582 | 128.3225450 | 37.5773941 | 823 | 343 | 51 | 170 | 8.4 |
| BOKSS | 2583 | 128.3247780 | 37.5794284 | 709 | 300 | 114 | 221 | 20.8 |
| BOKSS | Summary | | | | 581 | 165 | 194 | 15.9 |
| JS | 2584 | 128.5989230 | 37.4454180 | 1370 | 0 | 0 | 0 | 0.0 |
| JS | 2585 | 128.6002821 | 37.4502815 | 1218 | 553 | 152 | 193 | 15.4 |
| JS | 2586 | 128.6018278 | 37.4574335 | 919 | 805 | 299 | 190 | 20.4 |
| JS | 2587 | 128.6031841 | 37.4660038 | 639 | 959 | 280 | 187 | 16.3 |
| JS | 2571 | 128.6142804 | 37.4733674 | 390 | 1277 | 249 | 230 | 11.0 |
| JS | Average | | | | 3387 | 980 | 204 | 16.1 |

tions using the average slope direction, respectively. In these diagrams, time runs from top to bottom and altitude is from left
to right and the data is linearly interpolated in altitude.

Figure 5 shows an example of 24 hours of data (12KST from 21 to 22 Feb 2018) for Event 3 at JS. In this example, the
potential temperature shows a diurnal variation. Uniformity of potential temperature with altitude indicates a well mixed layer
in daylight hours (Feb 22 12-18 KST, marked WM) with a nocturnal stable inversion layer forms at night (Feb 21 21 KST -
Feb 22 9 KST, marked S). In this example, the transition from a well mixed to stable structure in the evening (line marked T-S)
occurs slower than the transition from stable to well mixed structure (line T-WM) in the morning ( 2 hours).

The example also shows variations in the co/cross slope flows. There is generally upslope flows at the top that appear to co-
incide with the presence of the stable thermal structure and a strong cross-flow (see red arrow linking the potential temperature
and WS10 co sub-plots). At lower altitudes, a downslope flows starts, in earnest, around Feb 21 16 KST, strengthens to about
Feb 22 0 KST and then weakens (see blue arrow). The cross-flows are weak at the lower levels due to the confining nature of
the narrow valley and the orientation of the winds. Not unexpectedly, the 1 min and the maximum winds show greater variation
and intensity. Given the events are multi-day and for conciseness, only the 10 min wind data were analyzed and presented
below.

Figure 6 shows Hovmueller diagrams for potential temperature and the co/cross 10 minute winds for all events and transects
investigated in this paper. A summary of the observations include:



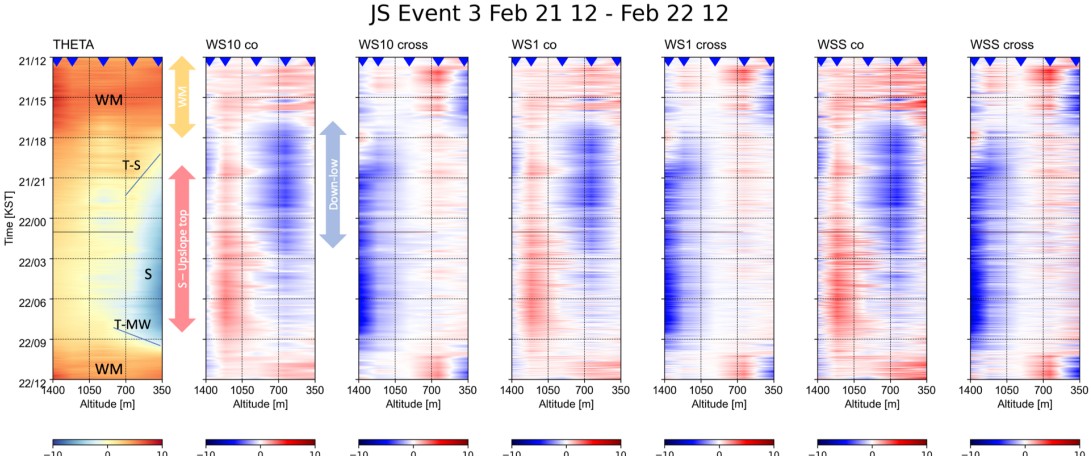

**Figure 5.** Hovmueller diagrams of potential temperature, co and cross slope winds (10 min, 1 min and max in 1 min) for Event 3 for the JS transect. The inverted triangles at the top of the plot indicate the altitude of the observations and the data is linearly interpolated between stations.

175   – For Event 1 (a,b,c), the potential temperature was the coldest of the three events.

  – For Event 1, the conditions were virtually the same for the entire two days indicating well-mixed conditions; whereas, diurnal patterns were evident in Events 2 and 3.

  – Events 2 and 3, exhibited well mixed conditions during the daytime hours and stable conditions at night, though the degree of stability (strength of the inverted potential temperature profile) was less or non-existent on the third night of
180   Event 3 (see sub-plot i).

  – The co/cross winds for each event and each transect exhibited a fine scale variation with time.

  – The top weather station at BOKX consistently show strong downslope flow.

  – Even though, BOKX and BOKSS were less than a kilometer apart, the co/cross wind patterns were different (compare sub-plots a to b, d to e and g to h) illustrating local effects. For example, at 850 m altitude, the co-slope winds were in
185   opposite directions. The cross-slope winds at BOKSS were consistently stronger than at BOKX.

  – For Events 2 and 3, for all transects and during stable conditions at night, the winds were generally light particularly at the lower altitudes.

  – For JS, for Events 2 and 3, the night-time flow pattern (with stable conditions) was complex with up and down slope flows at the upper and lower parts of the transect, respectively. During the day-time (well mixed conditions), strong
190   up-slope and cross-slope winds were evident (e.g. sub-plot f and i approximately between 12 to 16 KST).



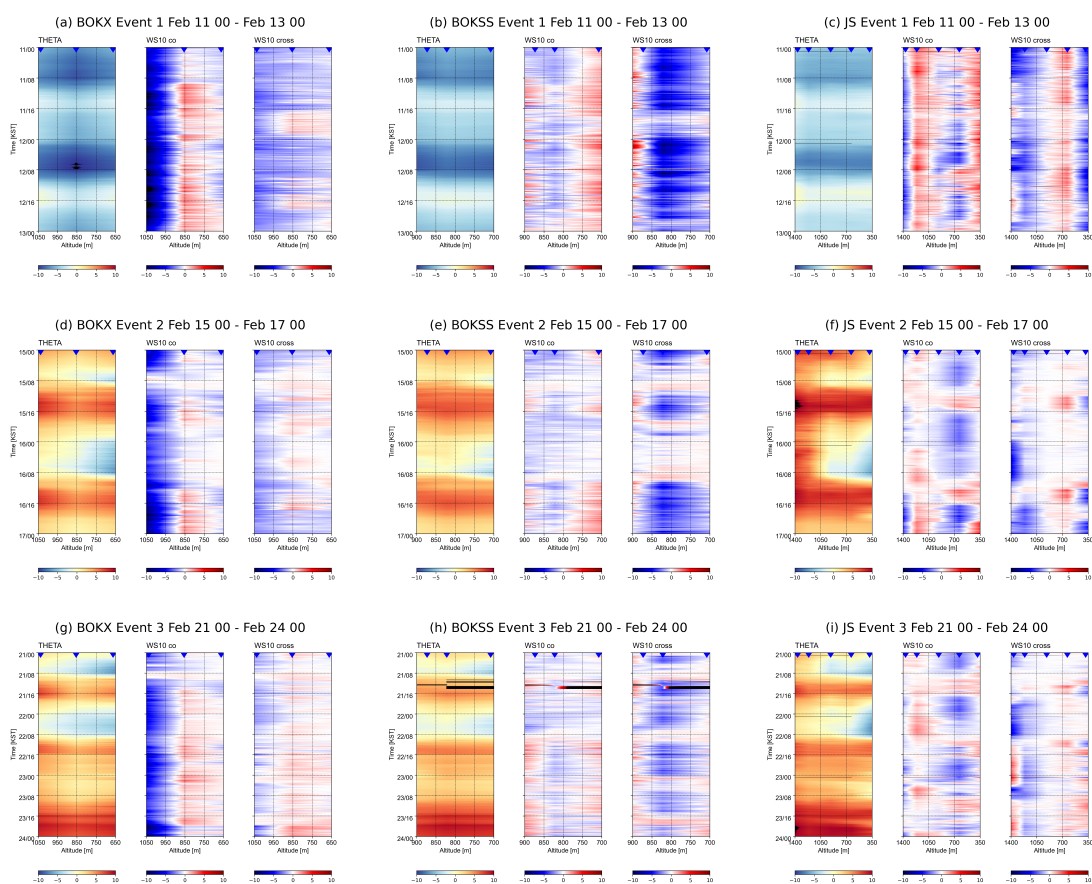

**Figure 6.** Hovmueller diagrams of potential temperature, co and cross slope 10 minute winds for the three events (row) and three transects (column) investigated. The first two cases (top two rows, labelled a, b, c, d, e and f) are for two days duration and the last case (g, h and i) is for three days. Note that the first two transects are physically about 1 km apart but the BOKX (ski cross) transect (a,d,g) has greater extent than the BOKSS (slope style) transect (b,e,h) and the altitude scales are different.

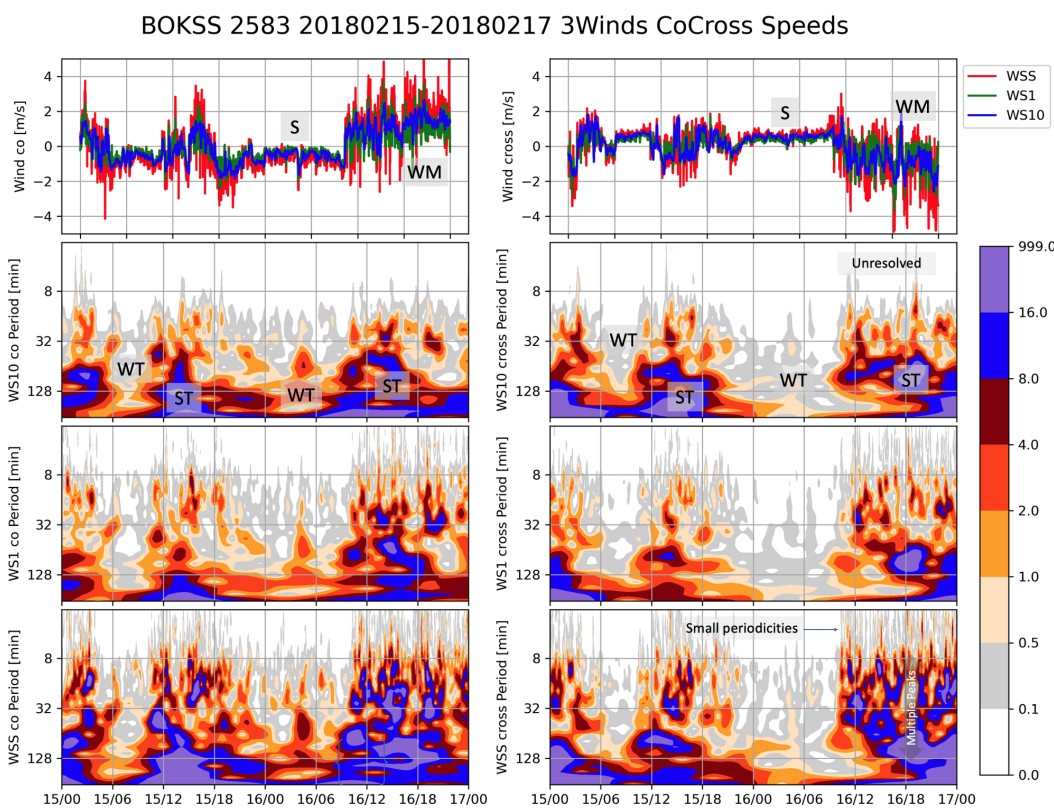

**Figure 7.** Wavelet Transform Analysis for station 2583 (mid station) for BOKSS transect for Event 1 (duration of 2 days). The top two plots show the co and cross slope wind speeds for all three wind types. The annotations S and WM indicate "stable" and "well mixed" winds. The lower three plots (top to bottom) are the corresponding wavelet transforms for the three wind types - WS10, WS1 and WSS, respectively. The annotations WT and ST indicate "weak" and "strong turbulence" regimes to aid interpretation for the reader. These graphs also show the benefit of higher resolution data combined with wavelet analysis to quantify the gusty winds. Note the annotations "unresolved" and "small periodicities", see text for more details.

- Of the three events, Event 1 exhibited the strongest winds.

- For Event 3, the day-time cross-winds were strongest on the first and second days (when competitions were cancelled) and weak on the third day. The co-slope winds shift from up to down slope during day-time on the last two days.

## 3.4 Wavelet Analysis

To first examine the fine scale variations in the wind patterns, a non-stationary wavelet transform analysis was performed (Torrence and Campo, 1998) to estimate the power spectrum of the co/cross winds. Figure 7 shows an example for all three wind observations (one minute resolution) for the lowest station (station number 2583) along the BOKSS transect for Event



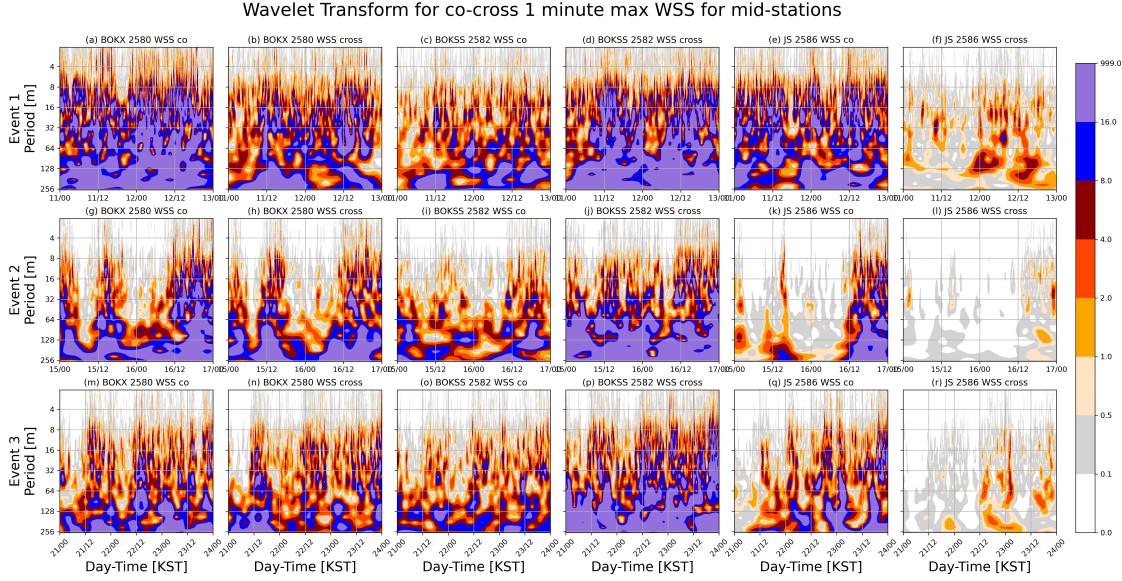

**Figure 8.** Wavelet Analysis Summary. The co/cross slope wavelet transforms for the mid stations (2580, 2582 and 2586) for the BOKX, BOKSS and JS transects are presented as adjacent pairs of images for the WSS wind type along a single row of plots for a single event. The three Events are presented by row. Note the first two events are 2 days in duration while the third event was for 3 days. See text for more details.

1. The left panels are for the co-slope winds and the right panels are for the cross-winds. This was prepared to illustrate the non-stationary diurnal pattern of the power spectrum and the impact of the different wind types. Diurnal patterns in the wind time series can been observed. Quiescent winds with little difference between the WS10, WS1 and WSS speeds (top two sub-figures) during the stable night-time hours (marked S) are observed, whereas there are significant variations during well-mixed day-time hours (marked WM) regardless of whether they are co or cross winds. This is particularly evident on the 16 Feb 2018.

The bottom three pairs of sub-figures in Figure 7 are wavelet transforms (abscissa is date-time and the ordinate is period and the colours are power). Not surprisingly but perhaps in a clearer fashion, the wavelet transform is consistent with the time series data and shows weak turbulent during the night-time and stronger turbulence during the day with most energy at larger periods/time scales. As expected, the WS10, wind (10 minute averages) are unable to resolve turbulence with periods below 20 minutes; whereas, the WS1 and WSS are able to capture the smaller periods with decreasing energy (Shannon, 1948). Also, as the spectra for the different wind observations are qualitatively similar, the WSS wind observations are subsequently used for further investigation.

As a summary, Figure 8 shows pairs (side by side) of co/cross WSS wind wavelet transform for the middle station (2580) for the 3 transects (column) and for each event (row). This provides a different perspective with the Hovmueller analysis but





for only a single station. The middle station was chosen as it corresponds roughly to the middle of the field of play. Many of the same features from the Hovmueller analysis are observed here with a different perspective and include:

- Event 1 was turbulent for the entire 2 days showing no night-time stable or quiescent period, whereas event 2 (a low wind event) and event 3 showed strong night-time stability with low turbulence.

- Not all power spectra extended to the smallest period resolvable. For 1 minute data, Shannon information theory indicates that 2 minutes, double the observation resolution, is the best that can be resolved.

- As with the Hovmueller analysis, for Event 1, even though BOKX and BOKSS were only a kilometre apart, the turbulence structure was quite different. The cross-slope turbulence was much stronger at BOKSS (compare b and d) whereas the co-slope turbulence was much more intense at BOKX (compare a and c).

- At JS, the cross-slope turbulence was much weaker than the co-slope turbulence in all events investigated likely due to the narrow confining valley.

- For Event 3, the turbulence on the second day (Feb 22 12-24 KST, sub-figure q and r), when events were cancelled, appears to be stronger than on the other two days of the event.

- The power spectra are not smooth but show a "lumpy" structure as a function of period/frequency.

### 3.5 Frequency Analysis

The fine scale variations were further quantitatively investigated using the Eigenvector (EV) frequency spectrum method (Marple, 1987; Johnson, 1982; Schmidt, 1986). This was chosen because of its precision resolving capabilities to extract significant frequencies from a time series. Similar to the multi-signal classification (MUSIC) technique, it does not attempt to conserve or retrieve the total power of the spectrum. While MUSIC weights the eigenvalues equally to compute the frequency spectrum, the EV method uses the eigenvalues as weights and claims to have greater capability to separate strong from weak signals. However, the power values at the significant peaks are similar to other power spectrum estimators (see Appendix Figure A1). The time series analysis assumes stationarity and hence the results are interpreted as averages over the analysis period. Details of the following analysis is described in the Appendix.

Figure 9 shows examples of EV frequency analysis for a two hour data segment. The figure uses the ten minute wind speed (WS10) time series data. The time series was scaled (mean of zero and variance of 1) for comparison purposes (Figure 9a). In this analysis, twenty eigenvectors (p) were a priori specified to estimate the frequency spectrum (see Appendix for explanation). The algorithm computes the eigenvalues of the covariance of the data and are shown in (Figure 9b). When the signal is strong, the eigenvalues are monotonic with eigen value number. The Akaike (AIC) and the Minimum Detection Length (MDL) theoretic information criteria claim to be able to determine the number of significant eigenvalues, separating signal from noise (Akaike, 1973; Wax and Kailath, 1985). However, depending on





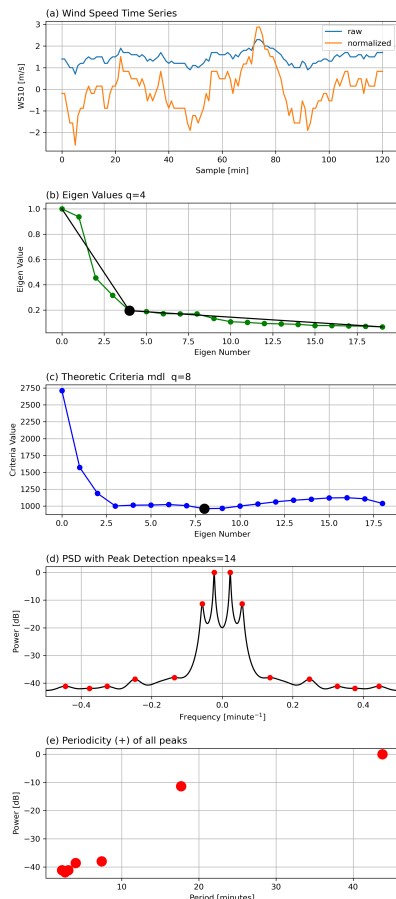

**Figure 9.** An eigen analysis frequency spectral estimator is applied to examine the periodic nature of the time series data. The sub-plots show: (a) the wind speed time series, as collected and normalized to a mean of zero and variance of 1; (b) the eigenvalues; (c the Minimum Detection Length theoretic information criteria used to identify the most significant eigen values (signal from noise) and (d) the frequency spectrum based on the most significant eigen values.





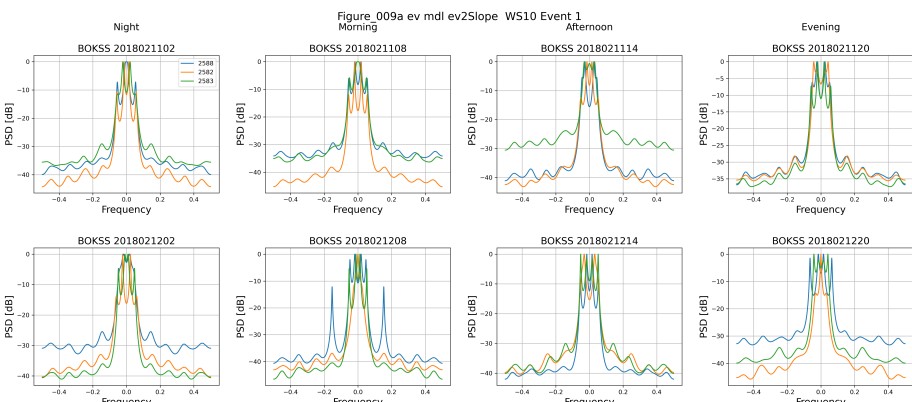

**Figure 10.** Example of a frequency spectra for Event 1 for the BOKSS transect.

the data, these criteria were not always well behaved, exhibited multiple or no minima. A new signal to noise separation technique eigenvalue based approach was developed. It exploited the observation that the eigenvalues for signal and noise (for strong signals) had two distinct slopes. This is termed the 2slope method and produced stable results and used in subsequent analysis(see appendix for details).

The method requires that the number of eigenvalues (p) be a priori specified. In general, the computed number of significant eigenvalues (q) was weakly dependent on p. Depending on the wind time series and the strength of the wind compared to noise, the number of significant eigenvalues (q) is generally around three to ten (Figure 9c). However, increasing p can create spurious peaks and so should be minimized (see appendix). The number of significant eigenvalues (q) is used to compute the frequency spectrum estimator. Given the sharpness of the spectral, a simple peak detection algorithm was developed and the peak frequencies and their power were extracted (Figure 9d,e).

Spectra as a function of the phase of the day, night (0-6 KST), morning (6-12 KST), afternoon(12 -18 KST) and evening (18-24 KST) for Event 1/BOKSS and for Event 3/JS transects are presented to shows examples of frequency spectra (Figures 10 and 11). The spectra are for the middle two hour segments of the phase of the day to investigate wind regime differences. The Hovmueller potential temperature analysis indicated that the valley was stable at night for Events 2 and 3 and lower turbulence may be expected. Throughout Event 1 and on the afternoons of Event 2 and 3, the atmosphere was well-mixed and in transition during the morning and evening phases.

Figures 10 and 11 show the spectra for each phase of the day. They show a consistency in the overall structure of the individual frequency spectra with generally 3 peaks at frequencies less than $0.2\ min^{-1}$. There is also general agreement that there is greater variability in intensity during the transition times (morning and evening) and less variability during night and afternoon.

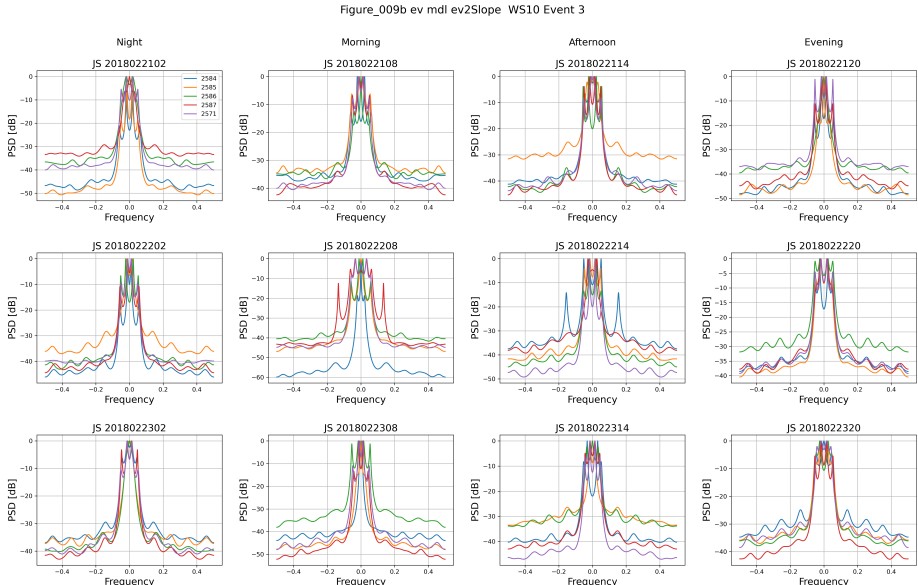

**Figure 11.** Example of a frequency spectra for Event 3 for the JS transect.

Figure 12a shows the distribution of the number of significant peaks from two hourly time segments for all venues and all events. Generally, the number of significant peaks in the distribution is between five and ten (see also Figure A3. In this figure, the distributions are expressed as periodicity which is the inverse of frequency. As the higher frequency peaks have generally lower power (Figure 12b,c), the power weighted number density distribution is computed (smoothed with a 5 point box filter) and is shown for various phases of the day (Figure 12d) and are generally similar. A peak analysis (for the entire day) indicates that the most frequent periods occur at around 22 and 52 minutes. The shorter one is sharp whereas the later one is broad and may have finer granularity. There are suggestions of longer periods at 74, 82 and 90 minutes though the data is sparse.

## 4  Discussion

Strong gusty winds were a significant issue during the Pyeongchang Winter Olympics and created changing inconsistent condition for the competitors but also for the organizers in deciding whether to conduct, to delay, to reschedule or even to change the rules of the competition. The competitions are conducted over a very small and limited spatial (tens and hundreds of meters) and temporal (tens of seconds) domain that is not normally served by national or private weather forecast services. The turbulence processes are not scientifically well understood, observed or analyzed, nor explicitly or implicitly well parameterized in numerical weather prediction models. In addition, in numerical weather prediction models, complex terrain is smoothed to prevent the generation of numerical noise and the structural details of the "field of play" such as the jumps or half-pipe are not captured



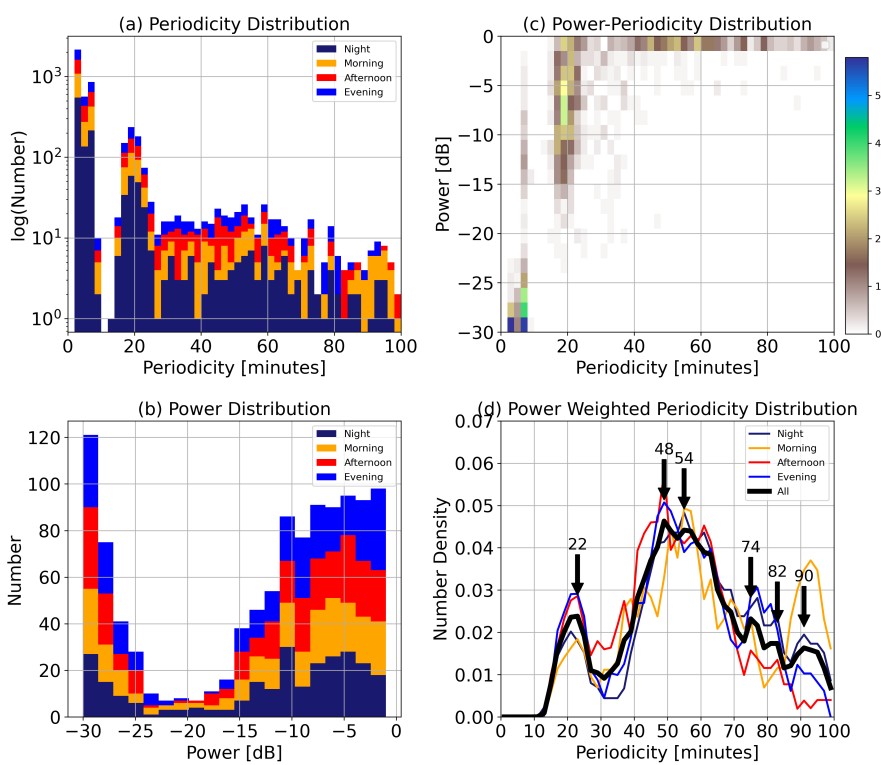

**Figure 12.** The frequency and power at the peaks in the spectra are extracted and analyzed by phase of day (middle 2 hour segment of phase period). The sub-plots shows: (a) a histogram of the the number of peaks found by periodicity (inverse of frequency). (b) a histogram of the peak powers and (c) the relationship between periodicity and power illustrating that the high frequency/low periodicity is weak and in the noise and (d) the power weighted periodicity number density distribution (normalized by total number). The arrows and numbers indicate automatically located peaks and periodicities in the distribution, respectively. Two distinctive dominant periods can be identified at 22 and 52 minutes (average location of the peaks at 48 and 54 minutes). The distributions in (d) have been smoothed with a 5 point box filter. There are suggestions of longer periods at 74, 82 and 90 minutes.





as boundary conditions. Hence, wind predictions and nowcasts primarily rely on observations and interpretation based on knowledge, experience and expertise much of which are developed during the Olympic effort. This investigation documented three events to extract nowcasting insights from an hindsight perspective and articulate the challenges in the context of the

observations, analyses, interpretation and the decisions.

*Impact of Synoptics* The large scale weather pattern shown by the time history of the lower tropospheric wind profile speed and direction (4) indicated a remarkable uniformity throughout the entire winter of west to north-westerly flow with some periodicity in wind speed magnitude though generally high ($> 7m/s$) at 700 mb which indicates strong influence of synoptics over local diurnal effects on the dynamics.

*Impact of Site* The Froude and Reynolds number analyses indicated similar conclusions and suggest that vortex shedding or wake turbulence were the dominant wind process affecting the venues which were in the lee side of the mountain. The use of $7m/s$ as a threshold between diurnal and mechanical effects may be too simple or too low as diurnal effects could be observed at higher values. The Froude or Reynolds numbers include consideration of atmospheric stability or viscosity. However, neither provided additional insight due to the limited variations in the atmospheric environment, locations studied and also due to the

complexities and realities of the terrain details.

The BOKX and BOKSS transects was located on an open broad slope and despite only being about one kilometre apart, the wind flow patterns were quite different. BOKX had stronger wind intensities than BOKSS. The BOKX had greater vertical extent, was closer to Peak B and the top station was also higher than that of BOKSS that contributed to the difference. The AWS stations were sited in both open and sheltered areas, for example, in the lee of a stand of trees (BOKSS, mid-station number

2582). The latter is not necessarily bad as it better represented the course conditions but must be taken into consideration when interpreting. Observation measurement error or frozen sensors could be a factor though visual inspection of the data did not reveal obvious data qualityissues. There were some occasions of bad data reported but, during the Olympic analysis period, they were sparse and were omitted or filled in by interpolation depending on the analysis requirements.

JS was higher and longer than BOK and was situated in a narrow sheltered avalanche chute that resulted in weaker winds

observed. There were significant differences in wind and turbulence intensities between the top to bottom stations due to the greater protection and narrowness of the valley at the lower altitudes. While gap winds may arise, such as those affecting Gangneung city just prior to the opening ceremonies, the relative wind and slope directions precluded this situation.

*Impact of Wind Types on Analysis Method* Three different winds were reported every minute for each station and included 10 minute averages, 1 minute averages and maximum 1 minute wind. Time series of wind speed showed the expected greater

variation with less averaging and can be used as crude indicator of the intensity of turbulent winds though validation is warranted to fully quantify its use. The Hovmueller analysis showed essentially the same patterns. The wavelet analysis was able to quantitatively show higher frequencies/lower periodicities than that using the WS1 and WSS measurements but the EV frequency spectrum estimation method investigated in this study was able to quantify the low frequency variations but not the higher ones. This may be attributed to the underlying white noise assumptions of the EV method and further development of

techniques to extract signal within a turbulent background are warranted but beyond the scope of this contribution. Though the





wavelet analysis indicated high frequency gusts, the one minute data is unable to capture the sub-minute scale of the gustiness of the wind that affected the competitors. This would require high response and high temporal resolution turbulence probes.

*Spatial/Temporal Nowcasting Challenge* The Hovmueller observations showed differences in wind patterns between BOKSS and BOKX that are only about 1 km apart. This is a challenge for normal weather forecast services to deliver products at such a scale and require dedicated forecasters per field of play field, as was done. Also, the local influences of the complex topography, stands of trees or artificial obstructions play critical roles and are not represented in the numerical weather prediction models. Hence, appropriate observations, and expertise to interpret them are needed.

The analysis also demonstrated that the transition from night time stable flow to well mixed day time flow can take two to four hours. These transitions are observed both in the morning and evening hours and need to be taken into consideration when using morning observations to decide on afternoon competition conditions (as in Event 1). However, in Event 1, the conditions were well mixed throughout the previous night and morning. It is not totally clear what was observed and reported by the venue forecasters and competition officials, but this analysis indicate the the wind conditions were similar or more intense conditions than the previous day. There may have been short lulls in the wind but given the observed periodicities in this hindsight analysis (Figure 12, observations of at least 20 minute duration should have been used to characterize the winds. In terms of user decision-making, at face value, the observations and nowcasts may be contributory but were not the determining factor in the decision to conduct the afternoon competition.

Not surprisingly, on the open slopes of BOK, the wind patterns can be dominated by either co or cross slope winds even if the fields of play are close together. Whereas at the JS, enclosed slopes, the cross slope winds are not dominant.

*Synoptic/Local Scale Linkage* The relationship between the 700 mb winds from the MOO sounding and the wind speeds (one hour averages at the time of the sounding) at the top of the each of the transects for the month of February was investigated and shown in Figure 13. Moderate to weak correlations are observed and are insufficient for precise decision making requirements supingport the need for in-situ monitoring.

*Wind Distributions* The wind speed distributions for the highest elevations for the three events are shown in Figure 14 and compared to those for the entire winter, February and Olympic periods. It is evident that the winds during Event 1 were by far the strongest winds throughout the Olympic and Winter periods for all transects particularly for station 2570 at the top of BOKSS. Event 2 was a priori selected as an example of low winds. The wind speeds for the two days show a bi-modal distribution implying that the low winds were anomalous and that the winds were strong throughout the competition. Event 3 while forecast to be strong were not that different and may have been even weaker than normal for the season.

*Hindsight Decision-Making Event 3* Figure 15 shows the distribution of wind speeds for the top stations for Event 3 on a daily basis. In Event 3, the first day of the event was the weakest in terms of wind strength and the decision to move competitions up to this day was appropriate. The winds on the second day were only slightly more intense than the winds on the third day. So, the decision to not conduct competitions on the second day was weakly substantiated.





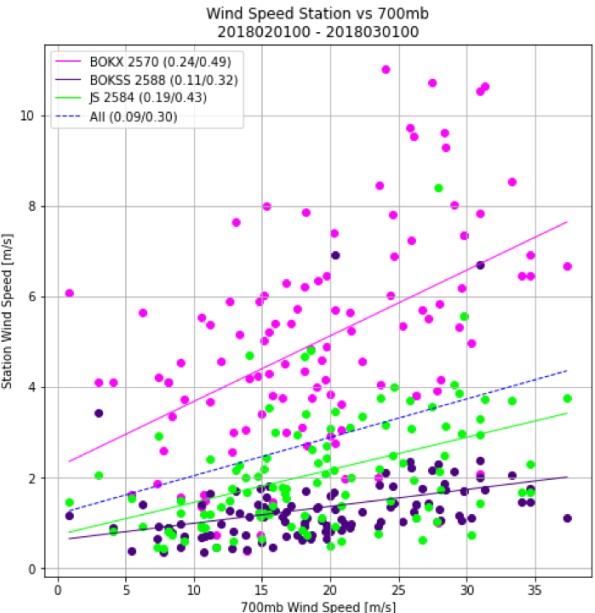

**Figure 13.** Scatterplot of the 700 mb versus top station wind speeds. The coefficient of determination ($r^2$) and correlation coefficients (r) are shown in the legend and are moderate to weak in strength.

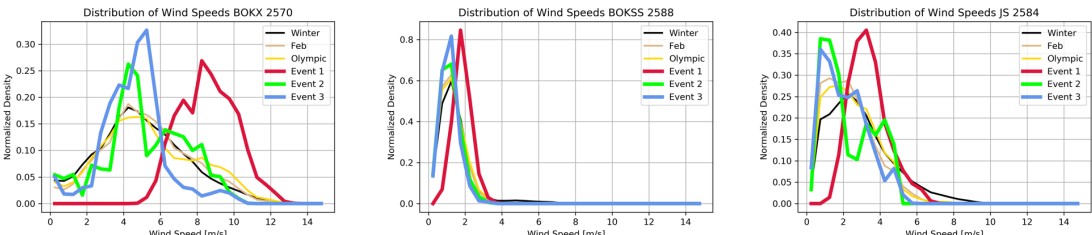

**Figure 14.** Distribution of wind speeds for the highest station at each transect showing that Event 1 was significantly stronger than other events investigated and for the entire winter.

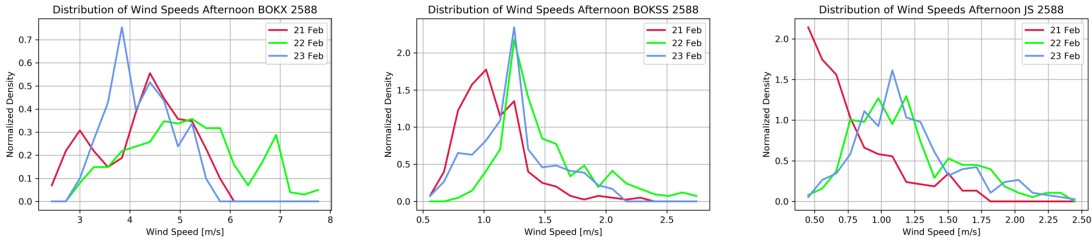

**Figure 15.** Distribution of wind speeds for the three days of Event 3. There were marginally stronger winds on the middle day when the competition was cancelled based on the forecast





## 5   Summary

Three wind events were analyzed to illustrate the nowcasting challenge faced by Olympic forecasters. The selected events
(Events 1 and 3) were arguably the most challenging to forecast/nowcast, but also chosen to illustrate the complexity of the
decision-making and limitations of the observations and knowledge. Two contrasting venues (Bokwang and Jeongson) and
three transects (ski-cross, slope style and downhill) were studied. The transects varied in length from about 700 to 2000 m that
indicate the fine scale challenge.

In complex terrain, the winds are influenced by many factors of different scale including the synoptic scale flow, thermal or
diurnal factors, the terrain and local obstructions. Previous results from the Vancouver 2010 Olympics showed that wind was
the worse predicted meteorological parameter (Huang et al., 2013).

The analysis documents the fine spatial and temporal resolution of the nowcast requirements that are much smaller than
that of normal nowcasting services. Nowcasting in complex terrain is generally beyond the mandate of national meteorological
services even in mountainous countries. Wind patterns and intensities can be quite different even though the transects were
less than a kilometre apart. While diurnal effects were observed, for the PyeongChang Olympic Games, the gusty winds were
generated by synoptic/mesoscale flows which were ubiquitous during the entire winter period.

The analysis showed that Event 1 had the strongest winds throughout the winter and well-mixed throughout the two days
including the night-time periods. The winds were similar on both days. The competition was cancelled on the first day but not
on the second day and illustrate that weather conditions partially contribute to the competition decision-making. In Event 3,
the weakest winds were observed on Day 1 of the three day event with Day 2 and 3 about the same. Competitions were moved
from Day 2 to Day 1 and 3. These examples illustrate the challenge but also the options that the competition organizers and
judges have available to them and that the impact of the forecasts and nowcast are a shared responsibility.

Hovmueller time domain, non-stationary wavelet and frequency spectrum analyses were applied to the three wind types
(10 minute, 1 minute and max within the minute) that reported every minute. All three methods showed periodic variations.
Quantitatively, the frequency spectral analysis showed several distinct periodic variations at 22 and 53 minutes (but have a range
of 7.5 to 60 minutes and perhaps more) suggesting the presence of lee-side vortex shedding. This was supported by a Froude and
Reynolds number analysis from a local upper air station. The wavelet analysis indicated variations down to 2 minutes at times,
which is the resolution of the information content (Shannon, 1948), indicating better high frequency resolving capabilities over
the eigen analysis frequency spectral technique. All estimation techniques have implicit assumptions and further investigation
is warranted to explore whether the technique could be improved taking into consideration the cascading nature of turbulence.

However, competitors faced sudden upslope and cross-wind gusts on a scale much smaller than 1 minute and 100 m that
caused them to perform badly, abandon or miss their jumps. This suggests that much higher frequency wind observations using
turbulence sensors are required. While 1 minute data may be novel to operational forecast monitoring networks, Olympic
organizers routinely deploy sonic anemometers recording 1 second data (Teakles et al., 2013) and depending on the analysis
method, even this may not be sufficient. The spatial variability suggests the deployment of Doppler lidars scanning or pointing
(staring) along and cross the slope would be most useful (Vasiljevic et al., 2017). Some of the wind flow may be caused by the





structure constructed to create the field of play (e.g. the jumps, rolls, ramps). Current forecast models are not able to resolve such features nor do they include the impacts/effects of the trees and other structures. Siting of the instruments to observe the field of play conditions and needed for nowcasting for the specificity of providing support of Olympic competitions are not

consistent with their use for the verification of the current generation of numerical weather prediction models

As many forecasters do not have experience and understanding of the scale and nature of winter complex terrain nowcasting, the path forward is to gain this knowledge is by a judicious interpretation of the numerical prediction models combined with investigative studies with the purpose of gaining experience, understanding and developing expertise. They do not have to be concerned about the impact of their nowcast as competition organizers have many options available to them and can

create alternatives taking into many non-meteorological factors including televised scheduling of other events at other venues. Organizers rely on their own global experiences conducting their events in winter complex terrain. Some sports are new to the Olympics and conducting competitions is also a growth experience for them. Interaction between the organizers and the forecasters to develop local weather knowledge, mutual understanding of roles and responsibilities will result in trust leading to better decision-making.

Scientifically, the next step is to develop greater understanding of the specific wind flow patterns and structures (i.e., nature and structure of vortex shedding). Future projects/venues will have their own challenges but the process to gain understanding will come from improvements in observations, understanding, sub-grid scale parameterization investigations in forecast models and perhaps even the use of high resolution (meter scale) computational fluid dynamic models.





# Appendix A

The time series analysis (i.e., the wind trace and Hovmueller diagrams) showed regular temporal variations in the wind speeds. The objective of frequency spectral analysis is to extract the dominant frequencies from a time series. Figure A1 shows an example of three classes of spectral analyses:(i) the classical correlogram, (ii) Burg"s auto-regressive maximum entropy and (iii) multi-signal eigen-analysis (Marple, 1987; Cokelaer and Hasch, 2017). The example is for a 2 hour period from 14-16KST 11 Feb 2018 (during Event 1) at station 2580 that is the mid-point on the BOKX transect. When the signal is strong,

the three techniques produce very similar results. However, the latter technique (EV, Johnson, 1982) was selected for further investigation as it has high resolving power which is defined as the ability to separate nearby adjacent peaks. Also, visual inspection of contiguous spectra showed that EV produced more stable or consistent spectra, fewer number of significant frequencies, fewer high frequency and weak peaks than the similar MUSIC technique (Johnson and DeGraaf, 1982). Eigen-analysis methods do not preserve the total power in the spectrum. However, the power at the estimated frequency peaks closely

match spectral power density with the other methods.

The EV technique first computes the eigenvalues of the auto-covariance of the data. The number of significant eigenvalues depends on the characteristic of the signal. The values decrease monotonically in magnitude with increasing eigenvalue number, as they are constructed in an iterative orthogonal process to explain the variance in the data. Generally, the first eigenvalues in the sequence are associated with signal and the last eigenvalues are associated with noise.

The algorithm can be configured with a known number of significant eigenvalues if the number of signals or peaks is known which is not the case here. If the number of significant signals are not a priori know, a threshold technique based on the magnitude of the eigenvalues or based on a theoretic information content approach may be used(Akaike Information Content or Minimum Detection Length, AIC or MDL; Akaike, 1973; Wax and Kailath, 1985). With both theoretic criteria studied (MDL and AIC), the separation of signal from noise space is identified as a minimum in value as a function of eigenvalue

number. However, minima were not always evident or multiple minima were present (Figure A2). It was found that the MDL criteria was better behaved (sharper minima and less noisy behaviour) confirming its claim (Wax an Kailath, 1985). However, neither were found to produce reliable results and a new technique to separate signal from noise was developed and described below.

In addition, a priori, the EV algorithm is configured with the total number of eigenvalues (p) to compute and it should be

greater than the number of expected significant eigenvalues or signals (q). If p is exactly set to the number of data points (upper limit), then the total number of eigenvalues will explain all the variance in the data (signal plus noise). Also, if p is large, the algorithm is able to resolve nearby frequency peaks but could also confound interpretation if the signal frequency is not steady. Hence, a procedure was required and developed to determine the minimum p to use.

The initial value of p was determined by a trial and error exploration of the MDL algorithm. Values of p were set from 10

to 60 which indicated that the minimum value of p was between 15 and 30 (there was little qualitative difference in results at higher values) and that the number of significant peaks (q) was between 5 and 15. With weak signals, the MDL theoretic criteria was unstable and unreliable.



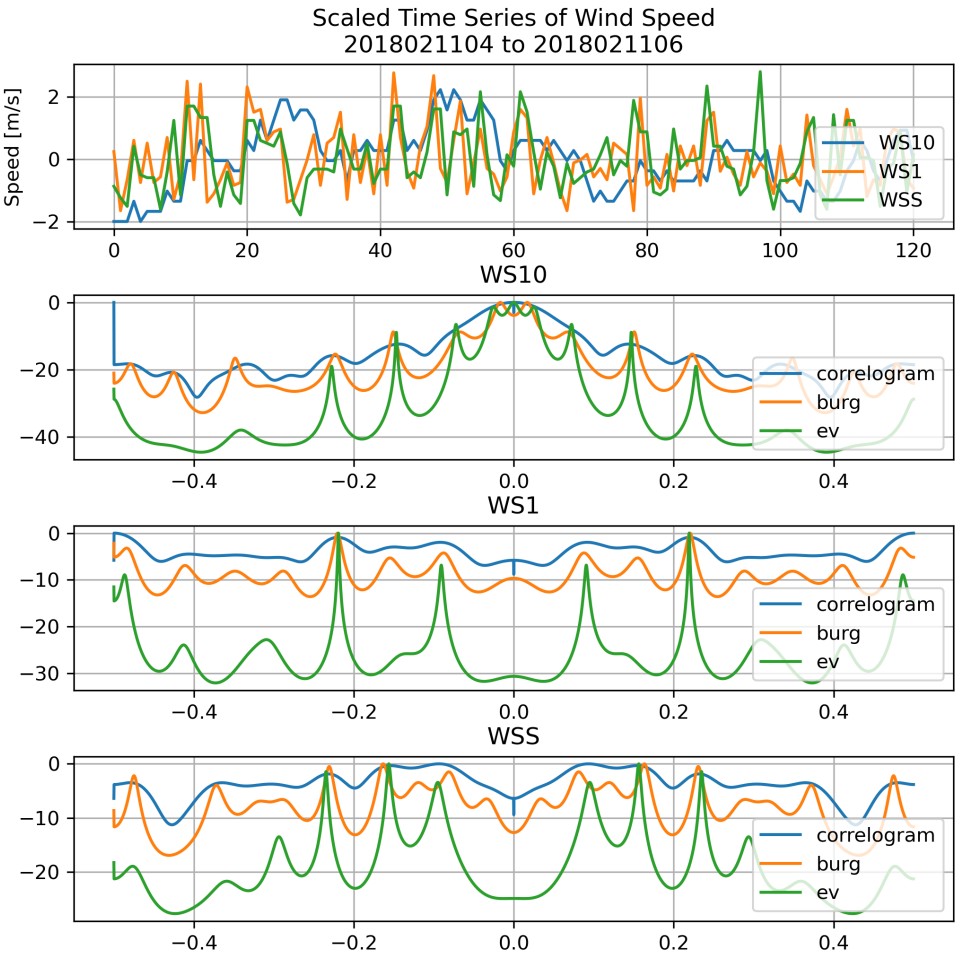

**Figure A1.** Example of spectral estimates of the three wind types showing the consistency of the significant frequencies when the signal is relatively strong.



Fig_App_002

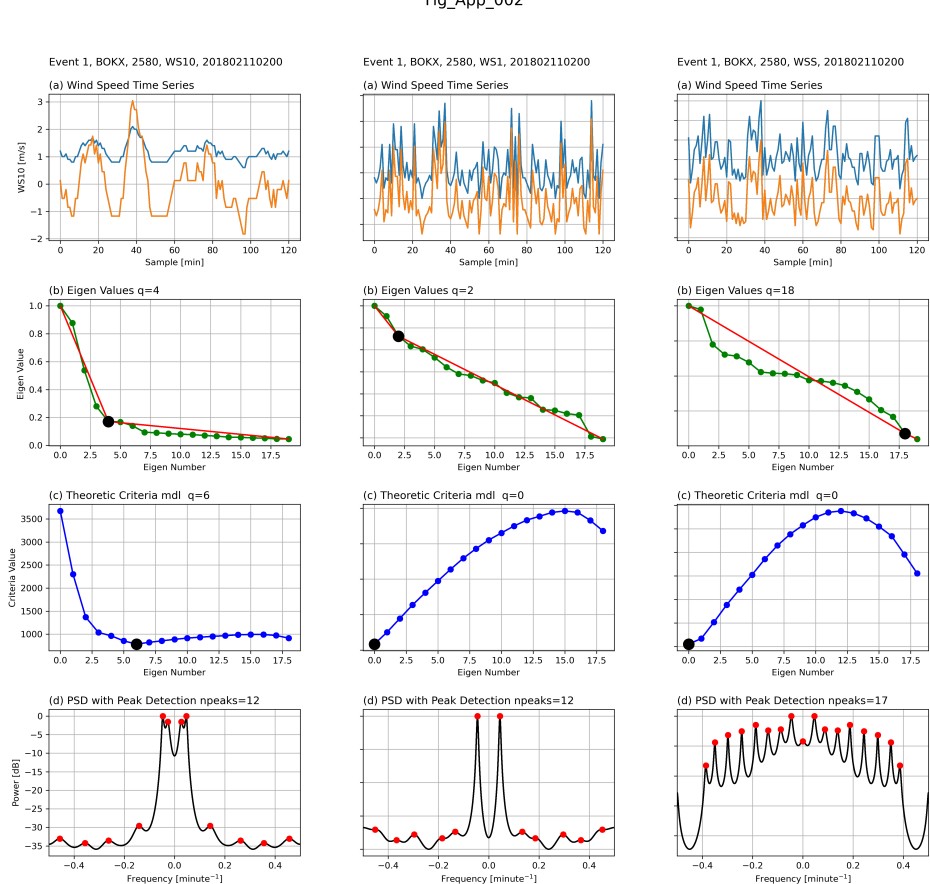

**Figure A2.** Examples of spectral estimates using the EV method with p=20 for the the three wind types is illustrated. The time series wind speed data (as measured and standardized to zero mean and unit variance) for the same time and station are shown on the top row. The eigenvalues are shown on the second row, along with the 2slope method (see text) to separate signal from noise. The theoretic information criteria (3rd row) patterns for good (left), marginal (middle) and poor (right) spectral estimates and the EV frequency spectrum estimate using the 2slope method (4th row) are shown. The markers on the EV frequency spectra indicate automatically detected peaks in the frequency spectrum. See text for further details.

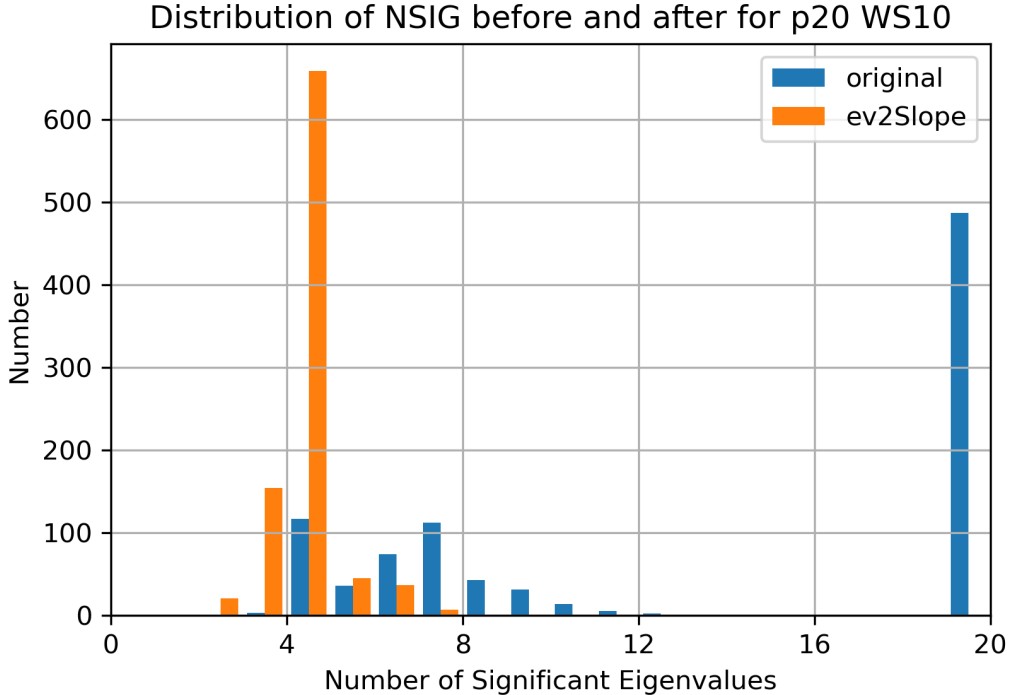

**Figure A3.** The impact of the MDL versus the 2Slope signal to noise separation methods are illustrated by distributions of the significant frequencies. The 2Slope technique detects a narrower distribution.

With sufficient signal to noise, the eigenvalues decrease rapidly in value with p and then flatten out forming a curve that can be modelled as two straight lines, one for signal and one for noise. The slopes of the lines are generally quite distinct and hence this is termed the "2Slope" method. The boundary between signal and noise is identified at the heel of the curve and computed at the intersection of the two lines (Figure A2b). The lines are anchored by the first and last values of the curve. Hence, p must be sufficiently large to reveal the noise part of the curve.

The 2Slope algorithm assumes that the first eigenvalue is signal ($p_0$) and the last eigenvalue ($p_{last}$) is noise. Intervening points ($p_{mid}$) are iterated through to define two line segments representing signal($p_0$-$p_{mid}$) and noise ($p_{last}$-$p_{mid}$). The difference between the curve and the lines are computed. The $p_{mid}$ with the smallest difference is selected as the point separating signal from noise.

For data with low signal to noise, consisting entirely of noise, the monotonic pattern of the eigenvalues may not be evident and may be "single sloped" or "flat" (Figure A2bc) or where the intersection point is beyond the limits of ($p_0$) and ($p_{l}ast$). Note the lack of minima in the AIC or MDL techniques for these cases (A2). Note also the automatic "threshold" technique was investigated where a factor (like 2) above the absolute minimum eigenvalue was a priori specified. But this proved problematic there was no information on the quality of fit to filter data consisting only of noise and was so it was not used.



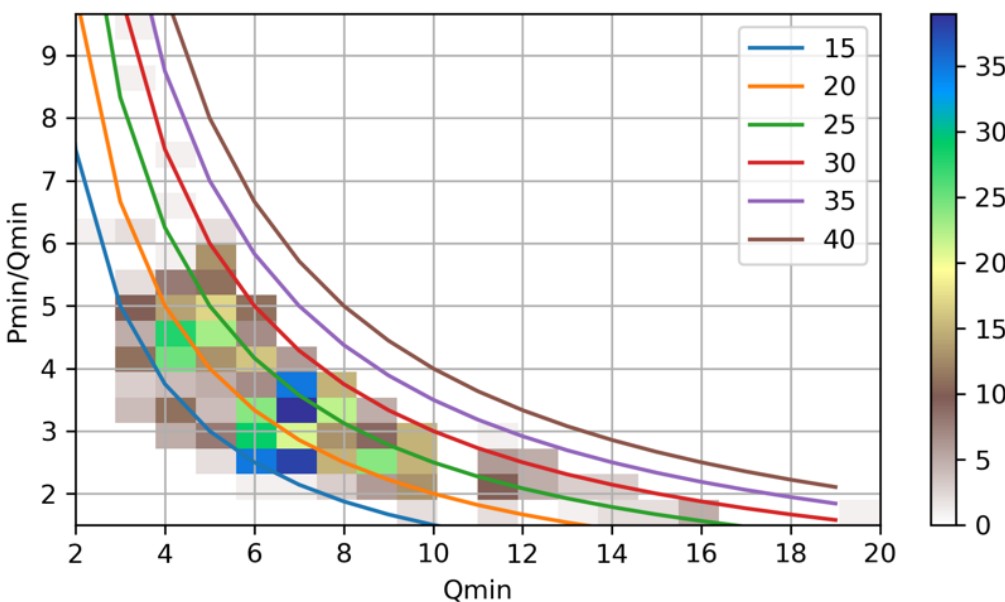

**Figure A4.** The ratio of the minimum number of eigenvalues to a priori use to the number of significant values are statistically related and is set to 20 for these data.

Figure 12 show frequency spectra for 6 hour data segments, for all three venues/transects, for all three cases and for all three wind types. The dominant frequencies are qualitatively similar for all wind types. However, for WS1 and WSS winds, about 15-20% of the cases indicated that no "signal" was found. The WS10 (10 minute running average) wind was almost always well behaved. The efficacy of the WS1 and WSS wind analyses require further investigation.

The frequency spectra for values of p from 10 to 60 and the statistics of number of significant eigenvalues (q) were examined for two hour segments, for all cases, venues and stations. Figure A3 shows the distribution of significant eigenvalues (q) illustrating that qualitatively it is about less than 5 and essentially independent of p. For each q found, the minimum number of eigenvalues ($p_{min}$) is determined as the minimum value of p beyond which q remains unchanged. Figure **??** shows that the ratio of $p_{min}$/q was a decreasing function of q. The solid lines indicate various values of p and from this, the value of twenty (20) bounds the results and was set for subsequent analysis.

Figure A5 shows an example of the EV analysis, partitioned in six hour segments by time or phase of day (night is 0 to 6 KST, morning is 6-12KST, afternoon is 12-18 KST and evening is 18-24KST) that may hint at greater variation in the number of significant signals during the morning and evening transition periods. The main text has further results.





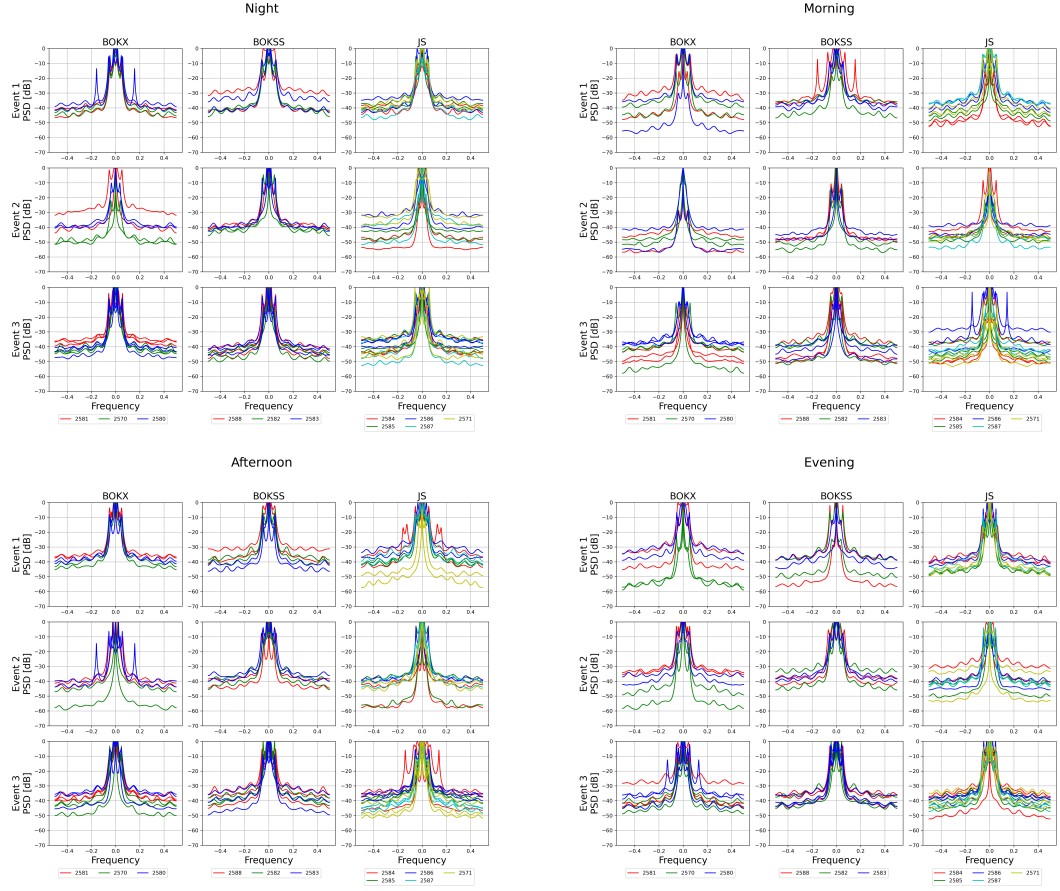

**Figure A5.** A summary of EV spectra for the three events by time of day. The day was segmented into 6 hour time periods (phases) to represent night, morning, afternoon and evening. Spectra from the middle two hours of each phase are shown to illustrate the diurnal pattern of the spectra.

*Acknowledgements.* The availability, use and quality of the AWS deployment, maintenance and data management by the technical staff of the Korea Meteorological Administration is greatly appreciated. The authors are greatly appreciative to the participants of the World Weather Research Programme Research Development Project and Forecast Demonstration Project, International Collaborative Experiments for Pyeongchang 2018 Olympic and Paralympic winter games (ICE-POP 2018), hosted by the Korea Meteorological Administration. G. Lee and K. Kim greatly appreciate the support from the National Research Foundation of Korea (NRF) grant funded by the Korea government 460 (MSIT) (No. 2021R1A4A1032646).





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
