# Peer review of "Measurement Report: Strong Valley Wind Events during the International Collaborative Experiment - PyeongChang 2018 Olympic and Paralympic Winter Games Project"

_Atmospheric Chemistry and Physics, 2021_

## Referee Comment (RC1)

**Review of "Measurement Report: Strong Valley Wind Events during the International Collaborative Experiment – PyeongChang 2018 Olympic and Paralympic Winter Games Project"**

Authors: P. Joe, G. Lee and K. Kim

The manuscript focuses on the analysis of strong gusty winds during the Olympic and Paralympic Winter Games of 2018. Strong gusty winds and their spatial divergence along the field of play can lead to unsafe and/or unfair conditions for athletes for example of the down-hill and half-pipe events. Accordingly, nowcasting is essential for the organizers of the events which need to decide if test runs or even the main event can take place or need to be rescheduled. Unfortunately, the scale or grid of forecasts is an order of magnitude bigger than the scale or size of the field of play making the decision for the organizers challenging.

Upper air analysis showed that the wind field was mainly synoptically forced by north-westerly winds. Near the surface, due to vortex shedding on the mountain ridges, these winds can lead to gusts and divergence of the wind field within the field of play. Further, local valley-wind systems can establish on the field of play which are not necessarily forcasted by common weather prediction models making the prediction of safe and fair events even harder.

The manuscript focuses on two venues and three events of which two were during the winter games which are case studies of 2-3 days for investigating the wind field rather than just gusty winds. Therefore up-to-date analysis techniques are used like wavelets, Hovmueller plots, and eigenvector frequency spectrum for determining the periodicity of gusts.

While this seems a valid approach to further investigate and evaluate the measurement network, the manuscript would benefit from the following:

- have a clear goal and narrative: I could not find a formulation of a clear goal or main objective like improving nowcasting. Also the mentioned objectives are quite broad and were not used to organize the mansucript. Accordingly, also the structure seems rather like a list of analysis tools than a cohesive study leading to improvement of our understanding. The analysis tools are chosen appropriately in my opinion, however, their results are not well organized nor well presented and most of the physical interpretation of the results is missing in my opinion. I highly recommend revising the manuscript such that a main goal is formulated and reached through a cohesive narrative.
- Figures: I can find mistakes, poor choice of organizing subplots, missing units, wrong colors, not good choice of color scales etc. in almost every figure (more details see below). Besides the figures themselves, the manuscript has too many figures. A selection would make sense in my eyes to have a more straight forward study instead of presenting all results. But if the manuscript remains a measurement report, maybe the amount of figures is appropriate.
- Analysis tools: Details are given below, but some techniques are not applied correctly (as far as I could see) their interpretation is incomplete or even incorrect in my eyes. As the main structure of this manuscript is missing, I stopped reading carefully after section 3.4. Also the Appendix seems very long and could be improved in a way that the frequency analysis using eigenvalues is better understood and also shows why this analysis tool is better or gives more insights than wavelets. So far I see not a discussion on this and I think one of the techniques is sufficient.
- Discussion: I could not find a real discussion in which publications are used to set the results of the manuscript into the context of our knowledge. If the manuscript remains a measurement report I would combine Discussion and Summary.
- Summary: I could not find what the novelties of the manuscript are, what the main outcome is or what the general implication for atmospheric science is. But this should be fine for a measurement report.

Due to my above mentioned points, I recommend major revisions.

Language:

Language use is correctly, but the manuscript needs more structure and a cohesive narrative to make it more reader friendly. Also for a measurement report some structure (like the formulated objectives) would be beneficial. Further, the manuscript would benefit from shorter and more precise sentences.

Title, L1, L14, L39, L41, L89, L97, L100, L126, Fig.7, L270, L355: You are using the phrase "strong gusty wind events", "gusty winds", "strong gusty winds" or similar versions of this phrase. Since this is the main topic of your manuscript I would highly recommend to keep wording the same: "strong gusty winds". Since the manuscript focuses on the wind conditions for different events, why not choosing a catchy title like "Measurement Report: Challenging winds during the International…."

Abstract

L5: "turbulence were" or "turbulence is" – Is turbulence itself influencing the athletes or the strong winds themselves?? Also do you actually have turbulence data? 1-min averages do not observe turbulence.

L6: "Three types of wind data" – I would argue that you use different wind statistics of one signal (10-min averages, and 1-min maximums of the 1-min signal)

L6-7: ...were reported every minute… → automatic weather stations with a 1-min resolution, right? What turbulence information did you get from the measurements?

L10-11: I am confused… do you mean: "Wavelet analysis was used for investigating turbulence while the method of eigenvalue analysis was utilized for frequency estimation of motions." (you specify in the next sentence how this indicated the frequency caused by vortex shedding)

Introduction

Overall the introduction is good and provides an introduction into the study. I only have a few comments or questions or recommendations:

Since the phrase gust is used multiple times, I would recommend to start with the definition of gusts and then lead to why or how they most likely affected winter athletes and then how it affected the PyeonChang Winter Olympic Games.

First paragraph (L14-17: competition was altered to provide safe conditions) is in contradiction to the second paragraph (L17-25: no safe conditions during women's slope style event). So was it just an attempt to provide safe conditions or did they not know better or was the competition altered after some events?

L19-21: what are the different ways? What is "small-scale nature of winds"? Why is the 1-2 minutes apart important? How does this lead to unfair competition?

L21: What are head-wind gusts?

L22: What is WSS? What does it stand for? I see that it is a reference, but I could not access the link (03.Nov, Austria) I also see WSS used later on for measurements. Please clarify.

L25-26: To avoid confusion, I would end the sentence after "...is conducted". Further, the phrase "race course" is used in L41 and L93 and should be adjusted

L27: "...extremely small by normal…" change to "...is a fraction of the scales used for normal operational forecasts." (assuming it is a fraction of the usual scales)

L27-30: You give an example of different "field of play" → it would be very beneficial to also have the scales for operational forecasts to give the examples more meaning…

L70: What is upper air observations? Radio sondes? Lidar measurements? I would at least mention what kind of measurement this is.

L75-78: Objectives are formulated: I did not find them again in the manuscript. Why is the manuscript not structured so it is easy to find the objectives (i)-(iv).

L79-81: Why is this not a summary of all sections? Instead the (i)-(iv) naming is used again which is very confusing

Project Background/Setup

Would it be possible to add a (small) map of Japan and location of the events and then the other included pictures of Figure 1?

Further comments to Figure 1 can be found below under the section "Figures".

L90: Could you add a marker for Peak B in the small plots?

L92: Abbreviations are introduced but I do not see where they are used afterwards. Delete if not used.

L94: Is it important to mention the avalanche chute? Seems quite irrelevant and is not mentioned again in the text

L100-104: This is results/interpretation/discussion which does not belong to this section!

L105: Since the events got labels and dates in a table, why not using this label "Event 1"? Same for the other events.

L112: add the section number/reference to the corresponding section

L112-113: I do not understand why this added information is interesting or relevant.

L114: "..are described elsewhere (Lee et al. 2021)." change to "...are described in Lee et al. (2021)."

L116: I would like to know what kind of upper air soundings or at least which temporal/spatial resolution the measurements had.

L117-118: so 1-min data and then 10-min averages were computed. Are the running averages overlapping? So I have 10-min averages every minute?

L118: Abbreviation "WSS": Since this is the max within one minute, I think a naming like $WS1_{max}$ would also make sense

L119: "...terrain, It should…" → "...terrain, it should…"

L119-120: The sentence is confusing, please revise.

Wind Time Series

L124-125: delete the sentence about other, but not used parameter or move to Section 2

L125: "...later two winds…" → use the introduced abbrevations WS1 and WSS. Further, of course WS1 and WSS have a higher fluctuation than WS10, because it is the mean of the signal acting like a filter. I do not see this as a major result or even mention worthy. But what I would suggest to introduce Figure 14 instead of Figure 2 here which actually gives statistics about the wind speed distribution of the events as well as overall during the winter of this year.

L127-129: A mean of a signal has less fluctuations than the signal itself. This paragraph and the corresponding Fig. 3 are redundant and do not provide any useful information.

Upper Air Analysis

L138: please provide dates when the period between Olympic and Paralympics was

L145-153: You use Reynolds numbers as indicator for mechanical turbulence as also other studies do. This statement is followed by "The interpretation of high Reynolds number is imprecise" making the before statement obsolete. Even the sentences afterwards don't make it clear what your real interpretation or conclusion is about Reynolds numbers. Please clearify if Reynolds number do indicate vortex shedding or wake turbulence or not.

Hovmueller Analysis

L157: From where are the potential temperature and wind speed measurements? I thought temperature measurements are not shown?

L158: "co-slope": I never heard that phrase and actually find it confusing in combination with cross-slope. I would suggest "along-slope" and "cross-slope".

L158-159: How were these components retrieved? How are you sure it is up-hill or down-hill? Did you also have information of the vertial wind speed? Was the rotation into the along-slope and cross-slope components 2-dimensional or 3-dimensional? Were the wind speed measurements perpendicular to the slope or aligned with gravitation? Please provide more information on this part

as you can only provide "real" down- and up-slope wind if the station was mounted as such. Otherwise the vertical wind component needs to be taken into account (3-dimensional rotation).

L159-160: essential to mention that altitude is decreasing from left to right! (maybe add a statement that it simulates the down-hill path an athlete would take

L170-171: "Not unexpectedly,…" → sentence redundant, delete

L171-172: I do not fully agree with the decision to proceed with 10-min averages. Especially since the events are just minutes apart and that you want to show the real gustiness of winds which can change rapidly, why choosing the 10-min averages??

L176-177: I see a diurnal pattern of rising and sinking temperature during all events, however, event2 and 3 differ from event 1 that they have cold-air pools, but also not every night. Please revise statement.

L178-180: Are nighttime conditions relevant?

L182: Any idea why?

L183-185: I would argue that the mentioned "local effect" is simply the different slope orientation of BOKX and BOKSS and not necessarily small-scale local motions like cold-air drainage or similar. The strength of along- and cross-slope flows of BOKX and BOKSS differ in strength, because the slopes have different orientation, but are part of the same valley. So when rotating the wind speed components into along- and cross-slope components into this framework the strength differs even if wind speed and direction is similar at both stations. I would look into spatial differences instead of comparing the cross-slope and along-slope winds to investigate local effects.

L186-187: I would interpret that cold-air drainage and pooling at the lower elevation lead to the low wind speeds.

L191: This is redundant and already mentioned, further, this is not part of the investigation, right?

L192-193: I do not agree, for example BOKSS cross-wind on event 1 are stronger than on event 2. Besides, are the observations for the along-slope winds relevant? If yes, for what?

Wavelet Analysis

Looking at the wavelet spectra I have some comments:
- the cone of influence is missing. Please add those to all your graphics
- WS10 can not be used for this analysis since it is a statistical metric from WS1. So analysing WS1 already contains every information which can be gained from this technique.
- Similar: the interpretation of the wavelet of WSS is complex and I actually do not know how to do it, since it is a maximum within a 1-min window, but the location of the maximum within the window is unknown. So analyzing the periodicity of a maximum, but its actual "time location" is unknown seems wrong to me.
- Accordingly, the only signal which can be investigated is most the 1-min averages.
- Looking at the spectrum: I am quite sure you did not rectify your wavelet power spectrum as presented by Liu et al. (2007). This is essential for this analysis! Please revise your analysis accordingly.
- I think when applying this correction, the results might be better comparable with the eigenvalue analysis.

I did not further read the text, because I guess it will be revised after the analysis is revised.

Liu, Y., San Liang, X., & Weisberg, R. H. (2007). Rectification of the Bias in the Wavelet Power Spectrum, *Journal of Atmospheric and Oceanic Technology*, *24*(12), 2093-2102. Retrieved Dec 2, 2021, from https://journals.ametsoc.org/view/journals/atot/24/12/2007jtecho511_1.xml

Frequency Analysis

Overall: I did not understand this analysis even when looking at the Appendix. Please make the analysis more clear, maybe even with simple example to understand how your detection of periodicity/frequency works. Further, there should be a way of averaging spectra and showing this

for each event instead of so many subfigures. Also the spectra look mirrored, so why showing both sides? What are negative frequencies?

Further comments:

L233-234: I do not understand the sentence about stationarity. You have to make sure there is no instationarity in your data or if so remove it by common techniques. So please provide information how this was assured.

L242: Any idea why multiple maxima/minima? Maybe also here some rectification/normalization is needed for analyzing the spectra?

L262: Why 2-hour segment?

L268: What do you mean with "finer granularity"?

L268: How can you have longer periods which are not even sampled twice within a 2-hour segment? I am confused.

Discussion

I am a big fan of combining results and discussion, so a narrative can be kept instead of separating results and discussion. The one paragraph simply a or multiple results. Next paragraph interpretation. Next paragraph discussion how this fits into the outcome of other studies. But this is up to each author how to approach this. Nevertheless I struggle with this section, because it is a summary not a discussion:

L270-280: thats part of the introduction/motivation for this study

L281-284: this paragraph wraps up the results of a previous section

L285-290: Where is this threshold from? Why should it be used? References?

L291-302: This is a description of the field sites and should be in Section 2

L304-312: The first three sentences are simply wrong (see comments before). The rest needs to be revised, because I am quite sure that Evs and Wavelet analysis should not differ this much! And if so, please discuss accordingly. Why is it here mentioned that gusts can not be resolved? I got the impression that the paper actually wanted to investigate those! Please comment on this.

L313-317: What would be the appropriate observations and what is needed to interpret them? Using wavelets or EVs to detect common frequencies of the last few hours?

L318-319: Where is this shown or investigated? Is that important for athletes?

L324: If the common periodicity of occurrence of gusts is 20min, how does a mean of 20min help detecting them? Averaging over this period might make them undetectable!

L327-328: This description of the hills need to be mention in section 2

L331: I would not bring a new figure with new results in the discussion! Seems like a complete new topic! Further, the correlation coefficients seem a little low to show a significant correlation between the parameter.

L332: "supingport" → "supporting"

L333-338: I would suggest to show Figure 14 way earlier (section 3.1), because it gives a good overview of the wind distributions during the different events! Further, the description of "bi-modal" is missleading, because the events show a different distribution, but I do not see any bi-modal distribution of one event!

Summary

Why are the suddenly citations in the summary? Anything up for discussion or being part of he introduction (like other studies already showed that…) should be mentioned earlier! Besides, the second last paragraph (L381-389) was confusing. Please clarify.

Figures (recommendations and comments)
- Figure 1: units are missing on color scale; I would use blue instead of gray for water; have a color scale without white; color brightness should increase or decrease with elevation; please add a bigger map of Japan; insert "Peak B" and "Peak J" in the subplots; what is SRTM03? Maybe add the synoptic flow as an arrow for each event.

- Figure 2: very sure the colors are wrong; maybe add rectangles for the events instead of arrows; or simply use Fig. 14 instead of this one; how is WSS wind direction defined?
- Figure 3: I would not include in the revised version; also colors are wrong
- Figure 4: quite sure the units of the color scale is wrong; also add units; use a gradient color scale with brightness of color increasing or decreasing; do not use white as a color!! What is the red line? → add some comments in the caption
- Figure 5: please reorganize date and time: 21 Feb 12:00 – 22 Feb 12:00; the 4th-8th plots are not discussed/presented: either delete or discuss in a more detailed fashion; add units to the color bar; the altitude decreasing from left to right is not intuitive → comment in the caption or add words like peak and valley to the plot
- Figure 6: "BOKX Event 1 Feb 11 00 – Feb 13 00" → "BOKX Event 1: 11 Feb 00:00 – 13 Feb 00:00" (otherwise very confusing!!!); super small figures; maybe choose a few and add the rest to the appendix; color scales need units!; what are the lines in subplot (h)?
- Figure 7: only use WS1; when the spectrum is rectified: use a linear scale, not an exponential!; Do not use white as a color! Add the cone of influence!
- Figure 8: see Fig. 7
- Figure 9: pretty sure WS1 is shown not WS10 (as written on the axis); Subplot (e) is not explained in the caption
- Figure 10 & 11: There should be a better way of presenting this… maybe averaged spectra or similar? Or something like Fig. 12?
- Figure 12: do not use white as a color!; why is there a "gap" in subplot (c)?
- Figure 13: would not include, do not see the gain in the figure
- Figure 15: maybe show as an opener showing what went wrong

I hope the detailed feedback does not discourage you! There is already a lot of great work done, it just needs some more work! If you can show what physical insight you gained and what you learned from this study, and what needs to be done for nowcasting of future events (maybe even on other sites), maybe this manuscript can even be turned into a publication instead of a measurement report (but that is up to the editor, not to me). Looking forward to the revised version!

---

## Author Comment (AC1)

**ICEPOP Valley Wind Paper**

**by**

**Paul Joe, Gyuwon Lee and Kwonil Kim**

**Combined Response to Reviewer 1 and Reviewer 2**

Thank you to both reviewers for their extensive and constructive comments. The lengths of the reviews is impressive. This response is is longer than the manuscript. This speaks to the interest and diligence of the reviewers. Your efforts are very much appreciated.

I provide some general comments and reactions regarding the overall message that I received and then respond to each of the comments on a point by point basis.

Reviewer 1 pointed to the lack of clearly stated objectives even though they are isolated in the penultimate paragraph of the introduction. The comments of lack of clarity of purpose of the eigenanalysis indicates that it was not clearly articulated or understood. Both reviewers pointed to the many small figures. Reviewer 1 commented on the possibility of this being a research contribution if improved. This was classified as a measurement report by the editor, and though I disagreed, I accept it to move forward.

Reflecting upon these comment, I felt that in hindsight, I had too many diverse points to make and that I was writing too tersely and making assumptions of the readers. Many of the comments are related to providing more context and explanation.

I adopted a follow the data approach in presenting the research. Sub-consciously, I wanted to clearly show in a tutorial fashion an analysis approach to provide a guide to forecasters using tools uncommon to forecasters including comprehensive examples on how to interpret the products. In hindsight, this diverted from "telling a story" approach. This also came out strongly from reviewer 2 who commented about "linking decisions with the data". Though, I felt that I did that, I did that in a very terse way and this indicated that I needed to provide more context and explanation.

I do not think that I made interpretation errors in spite of the minor errors and omissions in the figures. I think the discrepancy is with the notion of turbulence. This is indicated in the comment by reviewer 1 that "finer than 1 min sampling" is required for turbulence. I would agree that it is required to measure the full energy spectrum but I adhere to the L.F. Richardson and Komolgorov notion that atmospheric turbulence ranges from the planetary down to the Komolgorov scales. This indicates that I have to discuss this point to resolve the issues.

The advancement of "rectification" of wavelet transforms is novel to me and I very much appreciate the reference rather than propagate an obsolete analysis. However, in interpretation the wavelet spectrum, I actually took the rectification effect into account, without knowing about rectification. I was careful to only interpret the wavelet transform for location of peaks looking at power spectra in a local sense within octaves. I was not interested in the absolute energy as that will scale with the large scale flow (assuming Komolgorov inertial sub-range). I re-analyzed the data with rectification and while the images have changed, the interpretation remains the same.

So I do not see any interpretation issues.

Given the comments and the classification as a measurement report, I have decided to significantly reduce the scope of the objectives and the contents of the paper. I will focus on the nowcasting and process aspects.

As Reviewer 1 suggested, a new title to help focus the paper. I will focus on the Nowcasting Challenge of the Women's Slope Style Event (not final title), new analysis tools/forecast techniques and their limitations. I will not focus on reference examples (null or thermal case or with constricted terrain).

The most important points can be done by focusing on one event (Event 1) and one venue (BOKSS) and not three events and three venues. Then I can spend more space to provide more context/explanation. This will eliminate the figures with too many sub-plots. The number of figures may not change.

I will also drop the eigenanalysis. I may have not articulated my problems with the limited resolving capability of wavelet analysis clearly enough (i.e. poor resolving capability) but the eigenanalysis indicated that there were no closely spaced peaks and my concern was unfounded. While I felt that this was significant and answers a nagging question for me, it seems not important to the reviewers and so I will drop it as it is not needed to "tell the story". Publishing elsewhere will give the innovation that I found proper attention.

With the descoping, I will be able to focus on providing much more context which is the major message that I am receiving from the reviews. This will also limit the conclusions but also keep them simple and therefore understood easier.

Therefore, the suggestion by RC1 to change the title and the scope of the paper is accepted.

I actually do not see many disagreements but the comments help focus where I need to improve the manuscript.

**Point by Point Comments (Reviewer 1 followed by Reviewer 2)**

| | |
|---|---|
| While this seems a valid approach to further investigate and evaluate the measurement network, the manuscript would benefit from the following have a clear goal and narrative. I could not find a formulation of a clear goal or main objective like improving nowcasting. | See general comments above. |
| Also the mentioned objectives are quite broad and were not used to organize the mansucript. | There were many goals, I will descope and focus objectives. |
| Accordingly, also the structure seems rather like a list of analysis tools than a cohesive study leading to improvement of our understanding. The analysis tools are chosen appropriately in my opinion, however, their results are not well organized nor well presented and most of the physical interpretation of the results is missing in my opinion. | I adopted a tutorial-like, "follow the data" approach so that forecasters unfamiliar with the analysis technique could follow the logic of why they use these tools. |
| I highly recommend revising the manuscript such that a main goal is formulated and reached through a cohesive narrative. | These comments helped me re-shape the paper. I will limit the scope and limit the audience. |
| Figures: I can find mistakes, poor choice of organizing subplots, missing units, wrong colors, not good choice of color scales etc. in almost every figure (more details see below). Besides the figures themselves, the manuscript has too many figures. A selection would make sense in my eyes to have a more straight forward study instead of presenting all results. But if the manuscript remains a measurement report, maybe the amount of figures is appropriate. | Reduction of scope will reduce the use of so many subplots and figures.

In general, I follow the advice of "Tufte: Visual Explanation" for figure preparation and color scales but will review all figures during revision.

The number of figures may only reduce slightly but will not contain so many subplots. |

| | |
|---|---|
| Analysis tools: Details are given below, but some techniques are not applied correctly (as far asI could see ) their interpretation is incomplete or even incorrect in my eyes. As the main structure of this manuscript is missing, I stopped reading carefully after section 3.4.

Also the Appendix seems very long and could be improved in a way that the frequency analysis using eigenvalues is better understood and also shows why this analysis tool is better or gives more insights than wavelets. So far I see not a discussion on this and I think one of the techniques is sufficient. | Thank for the Liu, Liang and Weissberg reference. I was not aware of that paper and have revised the figures.

The frequency eigenanalysis was conducted to investigate the structure of the dominant frequencies in the data at high resolution. I mentioned the reason ("resolving power") in the manuscript, I assumed that this was known as it is text book material. I appreciate that it may not be broadly understood and I chose to be terse to be brief.

The wavelet technique requires the *a priori* specification of the frequency resolution and can potentially "hide" frequency peaks. However, the eigenanalysis did not show this in the data. Scientific insight and progress was made here but will be removed here and presented elsewhere. The suggestion to remove the eigenanalysis is accepted. |
| Discussion: I could not find a real discussion in which publications are used to set the results of the manuscript into the context of our knowledge. If the manuscript remains a measurement report I would combine Discussion and Summary. | Noted. There are many issues as the scope of the paper is formulated. Many issues are related to my personal experience with 5 Olympic projects - the nowcasting challenges, inadequacy of tools, use of observations, lack of focus on wind within ICEPOP, decision-making, etc.

Given the reviewer remarks, I have to descope.

De-scoping, simplification, restricting on just nowcasting will provide better context which I interpret the reviews as being the major weakness.

I don't really know what a Measurement Report is, I could not understand the distinction and why there is a distinction. If the Discussion is combined with the Summary, I don't know how to end the paper then. So I prefer to have a Summary. |

| | |
|---|---|
| Summary: I could not find what the novelties of the manuscript are, what the main outcome is or what the general implication for atmospheric science is. But this should be fine for a measurement report. | It was to document and demonstrate the challenge which is underestimated by organizers, forecasters.

ICEPOP was focussed on precipitation and microphysics. However, wind was the forecast issue during the Olympics and this is one of two manuscripts focussing on wind out of a planned 30-40 papers for the Special Issue. Precipitation occurred outside the Games period and had no impact on the Olympics. So this is the only nowcast paper in the special edition. There might be one or two others (not planned at the moment), but I am not aware.

Also, it will probably be the only paper focussed on the BOK venue for a variety of reasons some of which have already been mentioned.

As was pointed out, there is little documentation in the literature on nowcasting wind in complex terrain, on this microscale and how decisions are made. Teakles reference is one of the other ones but does not discuss the impact on decision making. My main goal was to call attention to this. |
| Due to my above mentioned points, I recommend major revisions. | OK |
| Language use is correctly, but the manuscript needs more structure and a cohesive narrative to make it more reader friendly. Also for a measurement report some structure (like the formulated objectives) would be beneficial. Further, the manuscript would benefit from shorter and more precise sentences. | Noted. Reducing scope should help this.

The objectives are stated but I adopted a "follow the data analysis" approach to lay out the bare facts before bringing them together to address the objectives in the discussion. I will adopt a "tell the story" approach. |
| Title, L1, L14, L39, L41, L89, L97, L100, L126, Fig.7, L270, L355: You are using the phrase "strong gusty wind events", "gusty winds", "strong gusty winds" or similar versions of this phrase. Since this is the main topic of your manuscript I would highly recommend to keep wording the same: "strong gusty winds". | Noted. Will review for this but not all gusty winds are strong enough to affect the competitors.

I chose these words carefully. They mean different things and I think I used them consistently but I will check. Reducing the scope will allow space to explain the issues more. |

| | |
|---|---|
| Since the manuscript focuses on the wind conditions for different events, why not choosing a catchy title like "Measurement Report: Challenging winds during the International…." | Thank you for suggestion.  This was a helpful suggestion in determining to reduce scope.

I am thinking of "Nowcasting challenges of gusty winds for the Women's Slope Style Event…." or something to be determined. |
| Abstract | |
| L5: "turbulence were" or "turbulence is" | "were" because it is referring to both vortex shedding and wake turbulence.

Both vortex shedding and wake turbulence are elements of turbulence.  There is a redundancy in the term wake turbulence. |
| Is turbulence itself influencing the athletes or the strong winds themselves??

Also do you actually have turbulence data? 1-min averages do not observe turbulence. | Both  affect the competitors but the fairness is determined by the gusty, intermittent strong winds.  If strong wind only, they would postpone the competition.  It was the intermittency that was creating the difficulty in decision making.

What is turbulence? There are several definitions..  It is non-laminar flow, it is unresolved fluctuations in the wind/velocity.

From a mathematical /NWP perspective, wind is a vector given by u,v,w which is then decomposed into $U + u'$, $V+v'$ where U,V is the "mean wind", dependent on the averaging (daily, hourly, minute) which are also related to spatial scales and $u',v'$ is the fluctuations and are sub-grid or unresolved components of the wind and parameterized by eddy dissipation rate which is a turbulent formulation.

From a atmospheric flow perspective (L.F. Richardson, Kolmogorov), scales of turbulence starts at the largest scales (Rossby or planetary scale) down to the Kolmogorov scale (cm or mm) and 1 min data can not capture the dissipation scales but certainly larger scales.  I assumed that this was understood but I will expand. |

| | |
|---|---|
| L6: "Three types of wind data" – I would argue that you use different wind statistics of one signal (10-min averages, and 1-min maximums of the 1-min signal) | I understand the reviewers perspective but given the multi-scale aspect of turbulence/winds, I argue that the difference in averaging capture different scales of the wind.

One interprets hourly, daily or yearly averaged winds very differently and so the word "type" seems more apropos but I am open to suggestions.

When I think of "statistics" of the wind, I think of probability distributions, perhaps even normal distributions and statistics of mean, mode, median, variance, kurtosis, … so I don't think is the right word. |
| L6-7: ...were reported every minute… → automatic weather stations with a 1-min resolution, right?

What turbulence information did you get from the measurements? | Yes, WXT520 is an automatic weather sensor. The manual indicates that reports of wind or other parameters can be configured at 1 second to 1 hour minute intervals presumably by simple averaging.

You get turbulence information at > 1 min (2 minutes according to Shannon) in temporal scale. |
| L10-11: I am confused… do you mean: "Wavelet analysis was used for investigating turbulence while the method of eigenvalue analysis was utilized for frequency estimation of motions." (you specify in the next sentence how this indicated the frequency caused by vortex shedding) | I will clean this up.  I did not say either point.

Wavelet analysis was used to investigate the wind or turbulent "power" spectrum  including the location of the dominant frequencies.

Due to it frequency resolving limitations, I used eigenanalysis to investigate the fine scale structure of the dominant frequencies (frequency spectrum without conservation of energy).

I was not interested in absolute value of power because it would just scale with the mean flow. I thought a reviewer would ask for the more advanced eigenanalysis. |
| | |
| Introduction | |
| | |
| Overall the introduction is good and provides an introduction into the study. I only have a few comments or questions or recommendations: | OK |
| | |
| Since the phrase gust is used multiple | Accepted, will do formally. I thought it was quite |

| | |
|---|---|
| times, I would recommend to start with the definition of gusts and then lead to why or how they most likely affected winter athletes and then how it affected the PyeonChang Winter Olympic Games. | evident.

I describe how gusty winds affect the competition unfairly and on the impact on the athlete in the first few paragraphs starting at line 21. |
| | |
| First paragraph (L14-17: competition was altered to provide safe conditions) is in contradiction to the second paragraph (L17-25: no safe conditions during women's slope style event). So was it just an attempt to provide safe conditions or did they not know better or was the competition altered after some events? | There is the "intent" and there is the "actual" situation AND then there is the changing of the competition rules.

This was discussed. Line 31+ |
| | |
| L19-21: what are the different ways? What is "small-scale nature of winds"? Why is the 1-2 minutes apart important? How does this lead to unfair competition? | I was reporting on the competition practice and this is just a fact.

The practice is to send competitors out every 1-2 minutes, not wait for 20 minutes or more as the competition has formal and informal time constraints for a variety of reasons - consistency of conditions, television schedule, fairness of competition…

They do not have infinite time to conduct the competition. |
| | |
| L21: What are head-wind gusts? | gusts in the face of the competitors |
| | |
| L22: What is WSS? What does it stand for? I see that it is a reference, but I could not access the link (03.Nov, Austria) I also see WSS used later on for measurements. Please clarify. | It was a reference to a "YouTube" video provided by the Olympic organization, but I see that it has been removed now probably because of the Beijing 2022 Games. I will have to remove this reference.

I found it again at….

https://olympics.com/en/video/women-s-slopestyle-final-snowboard-pyeongchang-2018-replays |
| | |
| L25-26: To avoid confusion, I would end the sentence after "...is conducted". Further, the phrase "race course" is used in L41 and L93 and should be adjusted | Noted and will adjust. |

| | |
|---|---|
| L27: "...extremely small by normal…" change to "...is a fraction of the scales used for normal operational forecasts." (assuming it is a fraction of the usual scales) | OK…will revise |
| L27-30: You give an example of different "field of play" → it would be very beneficial to also have the scales for operational forecasts to give the examples more meaning… | OK… can add this. |
| | |
| L70: What is upper air observations? Radio sondes? Lidar measurements? I would at least mention what kind of measurement this is. | Yes, radiosondes.  Default operational standard terminology.

Nothing else replaces radiosondes yet… |
| | |
| L75-78: Objectives are formulated: I did not find them again in the manuscript. Why is the manuscript not structured so it is easy to find the objectives (i)-(iv). | The objectives are located in the introduction already, a prominent place, labelled "objectives" and enumerated to highlight them.  IMHO they are clearly articulated.  So not sure what the reviewer wants and whether the suggestion is to provide the objectives earlier where context may not be evident. Will review introduction structure but comment is not clear and perhaps a "style" difference issue. |
| | |
| L79-81: Why is this not a summary of all sections? Instead the (i)-(iv) naming is used again which is very confusing | It is a summary of all sections.

They are not  the same as the objectives, new paragraph and re-numbering and should be evident.  Will revise with de-scoping of paper. |
| | |
| Project Background/Setup | |
| Would it be possible to add a (small) map of Japan and location of the events and then the other included pictures of Figure 1? | Yes.  Obviously needed as this is Korea.  Being part of a "special edition" and to reduce duplication and for brevity, I decided to NOT do an overview map.  The planned overview paper will have many more maps. |
| | |
| Further comments to Figure 1 can be found below under the section "Figures". | OK |
| | |

| | |
|---|---|
| L90: Could you add a marker for Peak B in the small plots? | Peak B is not visible in the small plots only in the upper right plot. I will indicate that it is off the map with an arrow.

I will revise this figure. I spent quite a bit of time to find maps showing the maps at the right scales. By dropping the JS venue, I will have space to add a larger scale image. |
| | |
| L92: Abbreviations are introduced but I do not see where they are used afterwards. Delete if not used. | I will clarify…I thought it was evident within the sentence as X and SS are adjectives of XC and SS for BOK. I use this thoroughout. But BOKX will be dropped. |
| | |
| L94: Is it important to mention the avalanche chute? Seems quite irrelevant and is not mentioned again in the text | It is extremely relevant, as it is a encapsulated by the terrain and hence a closed narrow slope compared to open slope of BOK. This is moot as I am removing this from the paper. |
| | |
| L100-104: This is results/interpretation/ discussion which does not belong to this section! | Will review, but it is not interpretation but the "objective/goal" and provides context for decision-making. My inclination is to leave it here but will review in revised manuscript. |
| | |
| L105: Since the events got labels and dates in a table, why not using this label "Event 1"? Same for the other events. | Good point, will do. |
| | |
| L112: add the section number/reference to the corresponding section | OK. Will add. |
| | |
| L112-113: I do not understand why this added information is interesting or relevant. | OK will remove. The focus of ICEPOP is on precipitation and microphysics. I wanted to highlight that wind was an issue. Only 2 paper out of about 40 are about wind events. |
| | |
| L114: "..are described elsewhere (Lee et al. 2021)." change to "...are described in Lee et al. (2021)." | Will revise. |
| | |
| L116: I would like to know what kind of | This is mentioned previously on line 70. I used a |

| | |
|---|---|
| upper air soundings or at least which temporal/spatial resolution the measurements had. | single and nearest radiosonde for this study. Overview paper will describe the extensive monitoring network. |
| | |
| L117-118: so 1-min data and then 10-min averages were computed. Are the running averages overlapping? So I have 10-min averages every minute? | Yes, this is what "running averages" mean. |
| | |
| L118: Abbreviation "WSS": Since this is the max within one minute, I think a naming like WS1max would also make sense | This is not my terminology but that used in the project and I expect the overview paper. WSS is kept for consistency with other ICEPOP papers and not something for me to change. |
| | |
| L119: "...terrain, It should…" → "...terrain, it should…" | Yes, this will be correct. |
| | |
| L119-120: The sentence is confusing, please revise. | Will review and revise for clarity. |
| | |
| Wind Time Series | |
| | |
| L124-125: delete the sentence about other, but not used parameter or move to Section 2 | OK will move to section 2 |
| | |
| L125: "...later two winds…" → use the introduced abbrevations WS1 and WSS. Further, of course WS1 and WSS have a higher fluctuation than WS10, because it is the mean of the signal acting like a filter. I do not see this as a major result or even mention worthy. But what I would suggest to introduce Figure 14 instead of Figure 2 here which actually gives statistics about the wind speed distribution of the events as well as overall during the winter of this year. | OK, will use WS1 and WS10.

re worth mentioning…not a major result at all but stating the obvious and expected.

re moving fig 14 to here… will consider in revised manuscript. I feel putting it later tells the story better. |
| | |
| L127-129: A mean of a signal has less | Agreed, this figure was to quantitatively and |

| | |
|---|---|
| fluctuations than the signal itself. This paragraph and the corresponding Fig. 3 are redundant and do not provide any useful information. | clearly show the nature / character of the signal for this case. I think it provides insight. Will add more context. |
| | |
| Upper Air Analysis | |
| | |
| L138: please provide dates when the period between Olympic and Paralympics was | This is mentioned previously on line 95. Paralympics is not relevant as none of the cases discussed were from that period of time. It will be in the overview paper. |
| | |
| L145-153: You use Reynolds numbers as indicator for mechanical turbulence as also other studies do. This statement is followed by "The interpretation of high Reynolds number is imprecise" making the before statement obsolete. Even the sentences afterwards don't make it clear what your real interpretation or conclusion is about Reynolds numbers. Please clearify if Reynolds number do indicate vortex shedding or wake turbulence or not. | OK, will review and make clear…. Simple Reynolds number analysis (as a forecast tool) indicates possibly both wake turbulence or vortex shedding.

The wavelet transform/eigenanalysis indicates vortex shedding. |
| | |
| Hovmueller Analysis | |
| | |
| L157: From where are the potential temperature and wind speed measurements? I thought temperature measurements are not shown? | This is mentioned in line 124-125. Temperature, pressure and humidity are not shown in the figure but are used throughout.

Simple temperature traces are not shown to reduce the number of figures. It isimplicitly included in potential temperature. Reviewer asked that this be moved to section 2. |
| | |
| | |
| L158: "co-slope": I never heard that phrase | This is unconventional but no incorrect. This is |

| | |
|---|---|
| and actually find it confusing in combination with cross-slope. I would suggest "along-slope" and "cross-slope". | common language in other fields such as weather radars with polarization diversity.

I was creating new terminology to try to be more precise and explicit. When writing, I thought carefully of the terminology and want to avoid "along" as it needs another adjective up/down or upslope/downslope and to avoid whether it was up/down in the gravity sense versus in a slope sense. It seemed to me to have more of a mathematical connotation if I defined it in the text. I will add additional text to explain why I am using this terminology. Reviewer 2 also had a comment about this.

I prefer to retain the terminology as it consistent but will clearly indicate in the text the rationale. |
| | |
| L158-159: (i) How were these components retrieved? (ii) How are you sure it is up-hill or down-hill? (iii) Did you also have information of the vertial wind speed? (iv) Was the rotation into the along-slope and cross-slope components 2-dimensional or 3-dimensional? (v) Were the wind speed | I completely agree with the reviewer and did not discuss this well enough in the manuscript It is deficiency of the observations in complex terrain but it is what the forecasters had to work with and the objective of this contribution was to investigate what could be done with the measurements and to highlight deficiencies |

| | |
|---|---|
| measurements perpendicular to the slope or aligned with gravitation?

Please provide more information on this part as you can only provide "real" down- and up-slope wind if the station was mounted as such.

Otherwise the vertical wind component needs to be taken into account (3-dimensional rotation). | (lessons learned). This is well known in the complex terrain community but is a bias in the forecasting community

This was a deficiency in ICEPOP and in the previous Olympic projects that I worked on (Vancouver and Sochi). Only a few 3D wind/ turbulence sensors were available and so vertical (in the gravity sense) wind measurements were not available.

This was noted for the current Beijing Olympics and more focus on wind measurements were made (including wind towers, doppler lidars). The sensors were measuring horizontal wind only and as the reviewer notes, the vertical wind is missing.

To answer the questions:
(i) the wind measurements are horizontal vectors (speed and direction),
(ii) the slope was determined from the average difference in horizontal location of the sensor site (lat,lon)
(iii) the dot and cross products of the wind vector with the unit slope vector was used to compute co/cross slope components.
(iv) two dimensional
(v) gravity |
| | |
| L159-160: essential to mention that altitude is decreasing from left to right! (maybe add a statement that it simulates the down-hill path an athlete would take | This is already mention on line 159-160 (altitude is left to right). I will add the word decreasing. |
| | |
| L170-171: "Not unexpectedly,…" → sentence redundant, delete | OK, will delete. I think stating the obvious assures the reader. |
| | |
| L171-172: I do not fully agree with the decision to proceed with 10-min averages. Especially since the events are just minutes apart and that you want to show the real gustiness of winds which can change rapidly, why choosing the 10-min averages?? | There is no problem with showing any of the results from the different wind types. The results/interpretation will all be the same, given the multi-day span of the event and the limitations of resolution for publication. |
| | |
| L176-177: I see a diurnal pattern of rising | 176-177 refer to event 1 as indicated |

| | |
|---|---|
| and sinking temperature during all events, however, event2 and 3 differ from event 1 that they have cold-air pools, but also not every night. Please revise statement. | 178-179 refer to event 2 and 3 and the authors point is there already |
| | |
| L178-180: Are nighttime conditions relevant? | yes, it indicates that it was a strong synoptically driven event and not localized thermal event and should provide an indication to the forecaster that it is a synoptically driven system and to base the forecast on that. |
| | |
| L182: Any idea why? | Yes, but that is a topic for another paper.

Initially, this paper was going to expand upon this but thought that I needed to solidify the frequency analysis.

I will remove this observation and save it for a future science contribution, as this is now categorized as a measurement report. |
| | |
| L183-185: I would argue that the mentioned "local effect" is simply the different slope orientation of BOKX and BOKSS and not necessarily small-scale local motions like cold-air drainage or similar. The strength of along- and cross-slope flows of BOKX and BOKSS differ in strength, because the slopes have different orientation, but are part of the same valley. So when rotating the wind speed components into along- and cross-slope components into this framework the strength differs even if wind speed and direction is similar at both stations. I would look into spatial differences instead of comparing the cross-slope and along-slope winds to investigate local effects. | I fully agree and this is one of my discussion points regarding nowcasting challenges.

Mountain alpine skiers know that conditions on one slope can differ from another slope. My experience is the nowcaster may not be skiers or familiar with the nuances of mountain weather and use the limited observations that they have access to to provide the same nowcast for all venues regardless of slope orientation.

I just present the observations here and tie it all together in the discussion. |
| | |
| L186-187: I would interpret that cold-air drainage and pooling at the lower elevation lead to the low wind speeds. | Yes, I fully agree. |
| | |

| | |
|---|---|
| L191: This is redundant and already mentioned, further, this is not part of the investigation, right? | Yes, just stating the obvious and which figure shows that, which may not be so obvious to the casual reader or target nowcaster audience. |
| | |
| L192-193: I do not agree, for example BOKSS cross-wind on event 1 are stronger than on event 2. Besides, are the observations for the along-slope winds relevant? If yes, for what? | As indicated, 192-193 refer to event 3 only.

In some events and venues, the issue was 'head' ('co', along slope) winds and for others it was the cross winds that affected the competitor and so both are important.  I discuss this elsewhere but will review manuscript to point this out. |
| | |
| Wavelet Analysis | Thank you for your insights. |
| | |
| Looking at the wavelet spectra I have some comments: | |
| • the cone of influence is missing. Please add those to all your graphics | OK, can do.  I purposely removed them because I was using many data points and the cone of influence was not that significant and left them out to simplify the already busy images |
| • WS10 can not be used for this analysis since it is a statistical metric from WS1. So analysing WS1 already contains every information which can be gained from this technique.
• Similar: the interpretation of the wavelet of WSS is complex and I actually do not | I view this from the signal processing perspective of digital filtering.  The sensor samples the wind at 1 second or better.  1 and 10 minute averages are produced using by the software in the sensor acquisition system which is essentially a boxcar filter where the 1 second samples are uniformly weighted. |

| | |
|---|---|
| know how to do it, since it is a maximum within a 1-min window, but the location of the maximum within the window is unknown. So analyzing the periodicity of a maximum, but its actual "time location" is unknown seems wrong to me. | However, there could be other filters (hamming, cosine windows applied)  and metrics used like the mode, the median instead of the mean of the distribution/samples.

It could also be a max value or peak detection filter.

These wind "types" were what was available and not normally used by forecasters and part of this investigation was to explore whether this has any value for nowcasting.

Indeed, it seems that it provides the forecaster with indiction of highly variable winds.  This is obvious but useful to have and the recommendation would be to keep this measurement as a poor man's turbulence indicator. |
| • Accordingly, the only signal which can be investigated is most the 1-min averages. | OK |
| • Looking at the spectrum: I am quite sure you did not rectify your wavelet power spectrum as presented by Liu et al. (2007). This is essential for this analysis! Please revise your analysis accordingly.

• I think when applying this correction, the results might be better comparable with the eigenvalue analysis.

I did not further read the text, because I guess it will be revised after the analysis is revised.

Liu, Y., San Liang, X., & Weisberg, R. H. (2007). Rectification of the Bias in the Wavelet Power Spectrum, Journal of Atmospheric and Oceanic Technology, 24(12), 2093-2102. Retrieved Dec 2, 2021, from https://journals.ametsoc.org/view/journals/atot/24/12/2007jtecho511_1.xml | I am very happy to receive this comment and has made this submission worthwhile.

I successfully published wavelet analysis, prior to 2007, and this issue was not known then.  I (and it seems like many others) accepted the explanation and interpretation by the originators and others and also from the perspective of the Kolmogorov -5/3 inertia spectrum that there should be more power at lower frequencies.

I have revised the analysis and the figures are evidently different but it has not changed my interpretation as I described in my general comments to this review. |
| Frequency Analysis

Overall: I did not understand this analysis even when looking at the Appendix. Please make the analysis more clear, maybe even with simple example to understand how your detection of periodicity/frequency works. Further, there should be a way of | Thank you for your comments.  As I am removing the "frequency spectrum eigenanalysis".  It was included to address a weakness (in my opinion) of the frequency resolving capability of the wavelet transform which is a priori specified and octave dependent. The eigenanalysis has the highest resolving |

| | |
|---|---|
| works. Further, there should be a way of averaging spectra and showing this for each event instead of so many subfigures. Also the spectra look mirrored, so why showing both sides? What are negative frequencies? | capability (according to Marple). However, the eigenanalysis results did not show close peaks. While this is good to know and reassuring, it can be removed from this manuscript without affecting the conclusions. The innovation will be published elsewhere. So my comments are moot. |
| **Further comments:** | |
| L233-234: I do not understand the sentence about stationarity. You have to make sure there is no instationarity in your data or if so remove it by common techniques. So please provide information how this was assured. | L233-234/262: I made the simplest assumption about stationarity - piecewise stationary of 2 hours (from inspection). |
| L242: Any idea why multiple maxima/ minima? Maybe also here some rectification/normalization is needed for analyzing the spectra? | L242: This is to be explored but I think it is because assumptions about the noise model is not satisfied for atmospheric turbulence (noise). |
| L262: Why 2-hour segment? | |
| L268: What do you mean with "finer granularity"? | See comment above. |
| L268: How can you have longer periods which are not even sampled twice within a 2-hour segment? I am confused. | L268: see previous comment regarding. |
| | The frequency analysis will be based totally on the wavelet transform. |
| | |
| Discussion | |
| | |
| I am a big fan of combining results and discussion, so a narrative can be kept instead of separating results and discussion. The one paragraph simply a or multiple results. Next paragraph interpretation. Next paragraph discussion how this fits into the outcome of other studies. But this is up to each author how to approach this. Nevertheless I struggle with this section, because it is a summary not a discussion: | I am strongly adverse to merging results and discussion. From high school lab reports to journal publications, I have been taught to keep them separate. There are the facts or results and there is the interpretation or discussion.

I can see that for short and linear investigation, merging may work but when one has to combine a few results and figures, as is the case here, a separate discussion section is needed.

When merged, I also find that the interpretation/ conclusions are all over the place and the summary has to duplicate all the conclusions rather than summarizing and pointing the direction ahead. |

| | |
|---|---|
| L270-280: thats part of the introduction/ motivation for this study | I will consider moving this up to introduction but I thought that the manuscript would benefit from a broader perspective before discussion of the data, as a whole, after laying out all the evidence. |
| L281-284: this paragraph wraps up the results of a previous section | OK. I will check for duplication. |
| L285-290: Where is this threshold from? Why should it be used? References? | The thresholds were mentioned earlier in   the manuscript (e.g L142 for 700 mb winds; L150 for Reynolds number; L143  for Froude number).

I can duplicate here again. |
| L291-302: This is a description of the field sites and should be in Section 2 | OK… |
| L304-312: The first three sentences are simply wrong (see comments before). The rest needs to be revised, because I am quite sure that Evs and Wavelet analysis should not differ this much! And if so, please discuss accordingly. Why is it here mentioned that gusts can not be resolved? I got the impression that the paper actually wanted to investigate those! Please comment on this. | I have revised the wavelet analysis using rectification.

The paper was to investigate whether the 1 minute data provided any indication of gusts which are short lived winds of the order of seconds, which can not be explicitly resolved due to the reporting of 1 min data by using the max wind or through extrapolation from larger scales. |
| L313-317: What would be the appropriate observations and what is needed to interpret them? Using wavelets or EVs to detect common frequencies of the last few hours? | L312 mention turbulence probes.

Later I mention Doppler Lidars (scanning over the field of play).

For nowcasting, tools like that presented Hovmoller and Wavelets plus conceptual models (for the moment) to interpret these tools/ products and obstacle scale CFD modelling (though not reasonable at the moment) |
| | |

| | |
|---|---|
| L318-319: Where is this shown or investigated? Is that important for athletes? | See Figure 5 and text describing figure 5. It is important for nowcasting and for competitions. Teakles et al had to provide nowcast during this transition time (10am-noon local time) and they show how the nocturnal cold pool breaks down (with cycle of ~7.5 min of up/down slope flows using 1 sec data) affecting the fairness of the competition.

In Vancouver, the competition was scheduled for television prime time in Europe where ski jumping is popular and this coincided with the transition. In PyeongChang, the example shows the transition duration of about 4 hours whereas Teakles showed a transition of 2 hours. This may be known within the mountain meteorology community but not so well known in the nowcasting community. I think providing more context will clarify this. |
| | |
| L324: If the common periodicity of occurrence of gusts is 20min, how does a mean of 20min help detecting them? Averaging over this period might make them undetectable! | I will clarify this. The decision to go or not to go was made in the morning, around 11am to conduct the competition later in the afternoon (1230-130 start). The information from this study indicates that one should base the decision not on a 20 min average but wind observations at 1 min (or better) for at least 20 minutes to see the periodicity. |
| L327-328: This description of the hills need to be mention in section 2 | The broad open terrain was mentions on L119. Here, I am trying to make a point for non-skiers that forecasting for one slope is not the same as for another slope for the same venue. Forecasters do not generally provide forecasts at this scale. I will clarify. |
| L331: I would not bring a new figure with new results in the discussion! Seems like a complete new topic! Further, the correlation | I had given consideration consideration whether to add a new figure or not in the discussion or whether this should be much earlier. |

| | |
|---|---|
| complete new topic! Further, the correlation coefficients seem a little low to show a significant correlation between the parameter. | whether this should be much earlier.

While it is not usual, there is nothing to preclude it either.

At the end, I thought the discussion flowed better here as it was advice/insight on how the decision-makers developed confidence with the forecasts and to the extent that they are able to alter the schedule. We found the forecasters painted too optimistic/pessimistic opinions and the organizers wanted the unblemished story - weak correlations included.

I will consider moving this to section 2. |
| L332: "supingport" → "supporting" | Thanks will revise |
| L333-338: I would suggest to show Figure 14 way earlier (section 3.1), because it gives a good overview of the wind distributions during the different events! Further, the description of "bimodal" is missleading, because the events show a different distribution, but I do not see any bimodal distribution of one event! | I will move Figure 14 earlier.

re bimodal comment… The L333-338 are poorly worded. Event 2 sentence is out of place. I will revise. As well, event 2 will no longer be discussed. |
| Summary | |
| Why are the suddenly citations in the summary? Anything up for discussion or being part of he introduction (like other studies already showed that…) should be mentioned earlier! Besides, the second last paragraph (L381-389) was confusing. Please clarify. | I was trying to avoid duplication but I can move text earlier into the introduction and repeat in summary that similar conclusions were reached in other studies.

I will clarify L381-389. We have found that this was/is a significant issue. |
| Figures (recommendations and comments) | |
| • Figure 1: units are missing on color scale; I would use blue instead of gray for water; | Will add blue instead of gray |
| | |
| | Will add units to describe the colour scale. |
| | |
| • have a color scale without white; | I disagree, white = bright = high elevation = snow capped mountains.

This is a standard colour scale for terrain. |
| • color brightness should increase or decrease with elevation; | Agree. White is the brightest colour and increase with elevation. |

| | |
|---|---|
| • please add a bigger map of Japan; insert "Peak B" and "Peak J" in the subplots; | I purposely did not include a larger scale map because of space, legibility and as this is part of a special edition and the overview paper will have the best large scale map. I did not want to create a potential conflict.

But the games were in Korea not Japan and this indicates that I should have a larger scale map.

Peak B and J are not within the domain of the sub-plots, increasing the domain and commensurate image did not contribute much and detracted from the seeing the terrain details.

However in the revision one of the sub-plots will be eliminated as well as the cross-section providing space for a larger scale map. |
| • what is SRTM03? | It is the name of the data set. Shuttle Radar Tomography Mission - 3 arc second. Will add a reference or remove. |
| • Maybe add the synoptic flow as an arrow for each event. | This already provided by Figure 4 and the user can use their favourite level.

I considered adding the 700mb flow but not sure how to capture the time history of the flow and then realized it was already in figure 4. |
| • Figure 2: very sure the colors are wrong; maybe add rectangles for the events instead of arrows; or simply use Fig. 14 instead of this one;

how is WSS wind direction defined? | Yes, you are correct. I made a last minute change to the colours for visibility and introduced an error in the legend

Will consider moving moving fig 14 and 15 here.

WSS wind direction is the direction at the time of max wind speed within 1 minute and defined on line 118. |
| • Figure 3: I would not include in the revised version; also colors are wrong | Agree. I was taking a tutorial approach in presenting the information.

JS events are no longer in the revised manuscript.

Yes, the legend is incorrect. |
| • Figure 4: quite sure the units of the color scale is wrong; also add units; use a gradient color | Will check and add units.

The direction colour scale uses the notion of |

| | |
|---|---|
| gradient color scale with brightness of color increasing or decreasing; do not use white as a color!! What is the red line? → add some comments in the caption | The direction colour scale uses the notion of white cold air coming from the north as white, red warmer coming from the south, prevailing westerlies as blue and green easterlies which is an intuitive scale as ascribed by Tufte: Visual Explanations |
| • Figure 5: please reorganize date and time: 21 Feb 12:00 – 22 Feb 12:00; the 4th-8th plots are not discussed/presented: either delete or discuss in a more detailed fashion; add units to the color bar; the altitude decreasing from left to right is not intuitive → comment in the caption or add words like peak and valley to the plot | OK…will re-organize time.

Will remove sub-plots.

Will add units

Altitude direction is intuitive as we read from left to right and the competitors go top to bottom. This is mentioned in the text but can add to caption. |
| • Figure 6: "BOKX Event 1 Feb 11 00 – Feb 13 00" → "BOKX Event 1: 11 Feb 00:00 – 13 | Sure, I checked HESS standards and will add colons. |
| Feb 00:00" (otherwise very confusing!!!); super small figures; maybe choose a few and add the rest to the appendix; color scales need units!; what are the lines in subplot (h)? | re notation…OK

re small figures…will revise and not adopt a tutorial style presentation.

Black lines in subplots are missing data. Will note in revision. |
| • Figure 7: only use WS1; when the spectrum is rectified: use a linear scale, not an exponential!; Do not use white as a color!

• Add the cone of influence! | The problem with a linear scale is the dynamic range of scales. Exponential is used to highlight this. This is a standard approach in the original and in subsequent wavelet transform papers. This is very common in engineering and in turbulence research (e.g. the Komolgorov inertial sub-range figure in many textbooks.)

I did not include the cone of influence because it did not provide much information and detracted from the figure. The data set is very long and only the bottom corners are affected. I can add in revision or discuss in test.

I have not problem with changing white in this figure. However, I kept it to be compatible with the original and other existing publications. |
| • Figure 8: see Fig. 7 | see previous comment |
| • Figure 9: pretty sure WS1 is shown not WS10 (as written on the axis); Subplot (e) is not explained in the caption | These figures will be removed in revised manuscript. |
| • Figure 10 & 11: There should be a better | |

| | |
|---|---|
| way of presenting this… maybe averaged spectra or similar? Or something like Fig. 12? | |
| • Figure 12: do not use white as a color!; why is there a "gap" in subplot (c)? | There is no gap, just not many points. This is shown in the power distribution in (d). |
| • Figure 13: would not include, do not see the gain in the figure | Forecasters look for correlations in preparing their predictions. This provides uncertainty or probabilistic information. This is one of more important figures. |
| • Figure 15: maybe show as an opener showing what went wrong | Not quite sure what the reviewer means but his event was included to show the dynamically changing decision-making environment as the organizers developed trust with forecasters and vice versa, the decision making process changed. This shows what actually happened and validated the decision-making to compress the events and move the events forward or backward.

I will consider highlighting Event 1 more in this style in revision. |
| I hope the detailed feedback does not discourage you! There is already a lot of great work done, it just needs some more work! If you can show what physical insight you gained and what you learned from this study, and what needs to be done for nowcasting of future events (maybe even on other sites), maybe this manuscript can even be turned into a publication instead of a measurement report (but that is up to the editor, not to me). Looking forward to the revised version! | Thank you for the encouragement. Much of the criticism was warranted and accepted. Your comments helped me to critically decide on the messages that I wanted to convey.

I adopted a "follow the data", "introduce new forecast tools" and a "tutorial" approach which I will suppress and adopt "tell the story" of event 1 with a follow-on event with respect to decision-making. |
| | |

Reviewer 2

| 1/ General comments | |
|---|---|

| | |
|---|---|
| The authors report results from different wind data analysis techniques applied to complex terrain (steep slope in a mountainous region) during and between Winter Olympic sports events. They intend to evaluate the rightfulness of decisions made regarding the cancellation and/or delay of said events. The title clearly reflects the article's content, which is relevant to the ACP publication. The strength of this article lies in the variety of approaches taken to study the wind at a high spatial and temporal resolution. Still, it contains many technical mistakes and could really use more proofreading. | Thank your for your review. They help change/re-focus my intention with this paper. |
| A critical comment to address in priority is the complete lack of literature references in the Discussion, part 4. Moreover, the publication could really take advantage of putting forward the difference between the decisions taken against the data available at that time. It seems that this is an important objective for the paper, but it is only mentioned briefly in the last part. | This comment is not clear to me.

I have some references in the discussion but most of the references were presented earlier. This is normally the case to me.

I tried to adopt a "follow the data" approach in presenting the materials. I did describe the decisions that were made of conducting the event even though the winds did not subside or change from the previous day in event 1. What is not possible is to know how the decision-makers trusted the forecasters, the information and what other pressures that they were under. |

| | |
|---|---|
| Another important concern is the lack of details about how the fine-scale terrain structure (presence of features like a half-pipe, trees, etc.) is addressed. The authors recognize that this is an issue for the type of analysis they conduct, but it doesn't seem to be considered in their analysis. | From the visual observations, it is clear that the fine scale details of the terrain and the obstacles play a big part of wind.  One can see the snow blowing and swirling from the obstacles.

The forecasters only had very limited meteorological information from which to make decisions.  The goal of this paper was to examine what could be done with this limited data with recommendations for future Olympics or similar micro-scale forecast applications (urban services for example).

The measurements were very limited and this was known prior to the Olympics and suggestions were made to improve them (including the loan of wind lidars for this venue). A paper such as this one to demonstrate what can or can not be done with the data is needed to provide formal recommendations for the future.

I will make this point more emphatically in the introduction. |
| | |
| 2/ Specific comments | |

| | |
|---|---|
| (l.125) I believe the paper would benefit from showing the temperature, pressure, and humidity values. Also, it matters to show the local or average slope angle and total change in terrain elevation when studying slope flows. | I started with plots of T, P, H and Wind but there were too many plots to show even when I plotted them as Hovmueller figures (which I produced but chose not to show).

More importantly, no significant statements/conclusions could be made as they basically showed diurnal, thermal and altitude effects. I realized that all this information is encapsulated in potential temperature which provides important physical insights and easier interpretation. Hence basic or advance plots of T,P and H were not presented. |
| (l.141) Can you justify using the 700 mb winds in the present context? | See Whiteman and Doran, 1993 which is referenced. I followed their lead as a wind to indicate the synoptic wind.
This is also commonly used by forecasters as the "steering level" for weather systems. |
| (l. 142) It could be useful to state what length scale was used to define the Froude number here. | I use height of the mountain. I will add. It could be valley to peak height but it did not make much difference. |
| The whole part 3, Wind Analysis, does not read easily. It should be more concise and lacks references for using the methods presented in a similar context. | OK…I will review this section with this in mind. |
| (l.230) missing a reference for MUSIC. | The reference is Marple and the others in line 228 as MUSIC Is a variant on frequency eigenanalysis. |

| | |
|---|---|
| (l.324) Determining the initial value of p by a "trial and error exploration" sounds like a rather weak reasoning and prevents the generalization of this approach. | I debated whether to use the terminology of "sensitivity analysis" instead of "trial and error".  I was trying to be provocative. |
| | This is how this and all these techniques seem to be formulated. For example, choose number of octave, number components per octave in wavelet analysis. |
| | Here p must be selected to be greater than the number of distinct frequencies, it can be chosen to be very large and hence it can be generalized. |
| | Nonetheless, how large is large and my appendix describes how I approached the problem which is not found in text books or in the literature when searching for guidance.  I think I provide a robust pragmatic approach and a contribution to this field, but this is moot as I am removing this in the revision. |
| (l.358) Please elaborate on the following statement: "The winds were similar on both days", i.e., in which aspect were they similar? | OK, thank you…will do.  In terms of intensity and wavelet transform pattern. |
| (l.445) What does "well behaved" mean in the present context. | I meant the noise decreased with increasing eigennumber, the expected behaviour compared to increasing with increasing eigennumber. |

| | |
|---|---|
| Overall, I feel that the hypothesis of a vortex shedding is only weakly substantiated, as it is hard to derive from just the Froude number and Reynolds number in such conditions. | I came to the conclusion of vortex shedding through the presence of distinct frequencies in the wavelet transform and eigenanalysis. The alternative would be un-distinct frequencies indicating wake turbulent like flow.

The Froude analysis indicated that we were not in a windward side block flow situation but the flow went over the mountain ridges. The Reynolds number indicated that we were not in a closed lee side vortex regime which left vortex shedding or wake turbulence flow modes in the lee of the mountains.

It is through deduction. It is not clear how else to interpret this intermittency except that at Komolgorov scales, the physical dissipation structures have been hypothesized to be ribbon-like and intermittent. But we are not at that scale and hence I conclude that vortex shedding is a possibility. I think obstacle scale modelling may help resolve this but I think a multi-doppler lidar campaign (see Perdagao, …) are designed to resolve this. I am happy to provide more context on this conclusion. |
| | |
| 3/ Technical comments | |

| | |
|---|---|
| (l.1-12) Abstract: The part of the abstract where the work in the paper is described should be written in the present tense with the passive voice as opposed to using passive voice with the past tense. Using past tense for describing the work in the article may imply that new results emerged in the literature and the results presented here are no longer valid. I.e., instead of: "For the other events, diurnal variations were observed with a stable atmosphere at night, well mixed in the afternoon and with 2-4 hour transition periods in the morning and evenings.", the author should use: "For the other events, diurnal variations are observed with a stable atmosphere at night, well mixed in the afternoon and with 2-4 hour transition periods in the morning and evenings." | Ok… in my 30+ year career of writing scientific documents of various types, including by professional proof readers for World Meteorological Organization official publications, I have never heard this.  Scientific documents were always written in third person past tense.

However, I am happy to learn something new.  Will review HESS guidelines, accept guidance from the editor and also try to revise as per the suggestion.  Thank you. |
| (l.14-15) Introduction: First Sentence can be rephrased so as not to be a repetition of the abstract. | Why? I thought it was accepted practice.

I spent some time coming up with the a good topic sentence.  The abstract is consistent with manuscript.

But, I will review and accept guidance from the editor.  I don't see the need to be inconsistent or rephrasing. |
| (l.27-28) "It is noteworthy that the field of." —> "It is noteworthy **to state** that the field of" | OK |
| (l.29-30) The author should use one format for presenting units throughout the paper. I.e., instead of "2 kilometers along the slope with less than 1km in vertical extent.", the following may be used: "**2 km** along the slope with less than **1 km** in vertical extent.". There should be a space with the magnitude (the numbers) and the units. A similar issue emerges at (l.85-86: 'meters' or 'm'. And in many instances, such as in l.183-184, l.6: "10 and 1 min averages plus 1-minute maximums". | OK will review for this consistency issue. |

| | |
|---|---|
| (l.79-81) The organization description should indicate in which sections the written procedure are handled. I.e., In section 2, the venues and fields of play, the selected events, and available observations are briefly described; in section 3, the results from various advanced analyses are presented, etc. The numerals used within parentheses do not correspond with section numbers, and each item should include a verb or should be formed in a way that none of the items have verbs. | OK… I can add more detail and directly reference section numbers. |
| (l.83-84) The author should either use "100 km x 100 km" or "100 km square". | OK… |
| (l.87) Table 1 and Table 2 are not introduced in the text. The first table the author mentions is Table 3. Before presenting the tables, the introduction to those tables should be given. | Table 1 is on line 117 and Table 2 is on line 96. The location of the tables was set by the HESS Latex system. I can try to fix this. |
| (l.105) What does "(see below)" indicate? A section, a table? It should be clearly marked. | OK…good point, I will replace this text |
| (l.114) "are described elsewhere (Lee et al., 2021)" —> "are described by Lee et al.(2021)". | OK… |
| (l.127) The correct suffix for singular nouns should be used, i.e., "Figure 3 show" —> "Figure 3 shows". Such mistakes occur throughout the manuscript and should be proofread for similar errors. Similar to "Details of the following analysis is described in the Appendix." in (l.234 and "The BOKX and BOKSS transects was located" in l.291. | OK…will proofread with this in mind. |
| (l.212-213) The following sentence should be made clear: "Many of the same features from the Hovmueller analysis are observed here with a different perspective and include:" | OK…will elaborate on what "many" refers to. |
| (l.216-217) Please provide a reference for this statement. | OK The reference to Shannon was in line 207 already but I can repeat the reference. |
| (l.219-220) "BOKSS (compare b and d)" and then "BOKX (compare a and c)": please indicate the figure label, i.e., "compare Fig. 6 panel b and panel d). | OK |
| (l.282) Does "(4)" refer to "Fig.4"? | Yes, corrected |

| | |
|---|---|
| (l.440) missing the word "as" after "problematic". | Thank you. |
| (l.449) missing the reference for the "Fig??". | Error on my part, this appendix will be removed. |
| (l.454) "main text has further results" is too general. It may be worth highlighting those results again in 1-2 sentences. | This is moot as this analysis will be removed. |
| Overall, many figures are too small or contain too much information for comfort (e.g., fig. 2, fig. 6, fig.9). | OK…descoping the analysis and the tutorial-style presentation of the results will correct this problem. |
| When they contain several graphs, it could be helpful to label each graph individually (a), b), c), etc.). Most pictures also lack a label for the color bar used (units, variables name?) | I will check all figures for labels and units.  I only did it for the figures that needed them. |
| Fig. 1 should show regions from the large to the small scale. Also, BOKX and BOKSS are missing from it. | OK…will add a large scale figure.

BOKX/BOKSS are marked along the coloured arrows.  Will make them more evident.  BOKSS map will be removed |
| Some acronyms are defined after being used (e.g., AWS at l.117). | Indeed, will correct… |
| Some words put in quotes, whereas a few more words could be used to define them more properly (e.g., "open" l.119) | Noted… |
| There are many instances of missing parenthesis (e.g., l.75, l.95, l.263, l.324, etc.) | I don't see the missing parentheses in 75, 95

263 is missing, 324 is missing

will check for others |
| What are "co-slopes" (e.g., l.158)? Do you mean along slopes? | I replied to this for Reviewer 1. |

| | |
|---|---|
| Some orthographic mistakes could be avoided with more in-depth proofreading. "Slope style" is written "slopestyle", (l.239) "eigen value" is "eigenvalue". I think "Hovmueller" is written "Hövmoller". | Thank you.

slopestyle are all slope style now

eigen value are all eigenvalue (moot)

Both Hovmöller or Hovmueller are correct

all changed to Hovmöller |
| In Table 2, "For comparison" is a bit abstract. I would prefer "for reference". | That case has been removed. |
| What is the red line in Fig. 4 standing for? | The description of the red lines were inadvertently omitted in the caption but were included in the text. The caption has been rewritten. |
| Table 3 is introduced on l.88, but it only showed eight pages later. | This is a latex formatting issue and I try to move the tables but may have to leave it to the HESS latex gurus to correct. |
| In Fig.5, the variable THETA, most likely the potential temperature, should be precisely defined. | Thank you. It is now defined in the text. |
| In Fig.7, the horizontal axis is inconsistent between the top and bottom figures. | Noted. The grid lines are correct but the plotting package defaults need to be overridden. This will be changed |
| It is hard to distinguish each curve in Fig 14, plus the title overlaps the figure. | I do not see the titles overlapping the figures.

In colour, the different lines are evident. |
| There is no description of Fig. 9e in the caption. | Will revise. |
| The titles fig_009a or b in Fig.10 and Fig.11 should not be left there. | Thank you, will remove… |
| Figure 1 Caption: 100km x 100km —> 100 km x 100 km | OK |

| | |
|---|---|
| Table 1 and Table 3: Units for the Latitude and Longitude unit (degrees) should be given even if common knowledge. | Yes, thank you |
| Table 1: Year information for rows 1, 2, and 3 should be given. | OK…for Table 2 |
| Table 2: Full stops (punctuation mark) should be avoided unless in a complete sentence. | Will edit for complete sentences. |
| The format in references is inconsistent (e.g., l.509, space between initials needed). | I had checked the HESS web site and they show no spaces between initials. I will let HESS proofreaders adjudicate at the appropriate time. |
| Throughout the manuscript: Table captions are not descriptive. They should be as descriptive as the figure captions in the manuscript. | There are no captions just titles. I will provide captions. |
| Throughout the manuscript: 'Figure' should be abbreviated x —> Fig. x instead of tables. | Abbreviation is an option. I prefer to use full text. |
| Throughout the manuscript: The months should not be abbreviated. Feb —> February. | OK… I will adopt an all numeric date to address this comment. It is driven by column spacing in Table 1. |
| Throughout the manuscript: Parentheses with long descriptions should be avoided within the text for clarity. | I will review for this. Descoping and focussing on fewer objective should clean this up. |
| | |